# Continuous observations from horizontally pointing lidar, weather parameters, and PM$_{2.5}$: a pre-deployment assessment for monitoring radioactive dust in Fukushima, Japan

Nofel Lagrosas[1,2*], Kosuke Okubo[1], Hitoshi Irie[2], Yutaka Matsumi[3], Tomoki Nakayama[4], Yutaka Sugita[5], Takashi Okada[5], and Tatsuo Shiina[1]

[1]Graduate School of Science and Engineering, Chiba University, Chiba, 263-8522 Japan
[2*]School of Engineering, Kyushu University, Fukuoka, 819-0395 Japan (*current afilliation* )
[3]Center for Environmental Remote Sensing, Chiba University, Chiba, 263-8522 Japan
[4]Solar-Terrestrial Environment Laboratory, Nagoya University, Nagoya, 456-0000 Japan
[5]Graduate School of Fisheries and Environmental Sciences, Faculty of Environmental Science, Nagasaki University, Nagasaki, 853-0000, Japan
[6]Japan Atomic Energy Agency, Ibaraki, 319-1194 Japan

**Correspondence:** Nofel Lagrosas (nofel@civil.kyushu-u.ac.jp)

**Abstract.** A horizontally pointing lidar is planned for deployment with other instruments in Fukushima, Japan, to continuously monitor and characterize the optical properties of radioactive aerosols and dust in an uninhabited area. Prior to installation, the performance of the lidar is tested at Chiba University. Data from the continuous operation of the lidar from August 2021 to February 2022 are analyzed for extinction and volume linear depolarization ratio. These are compared with the weather sensor and particulate matter (PM$_{2.5}$) measurements to quantify the relationships between atmospheric conditions and optical properties of near-ground aerosols. The results show that lidar data's extinction coefficient and depolarization ratio can have a quantifiable relationship with relative humidity (RH), absolute humidity, rain rate, wind speed, wind direction, and PM$_{2.5}$ concentration. Analysis of the seven-month data shows that the optical properties of aerosol and dust depend on the combined effects of the weather parameters. An increase in RH or PM$_{2.5}$ concentration does not imply an increase in radioactive aerosols. The average extinction coefficient and depolarization ratio of aerosols and dust originating from the land and ocean show different values and opposing trends which can aid in determining the occurrence of ground-based radioactive dust and aerosols. The information obtained from analyzing the interrelationship among lidar, weather parameters, and PM$_{2.5}$ concentration is essential in assessing the occurrence of radioactive aerosols and characterizing local aerosol-weather relationships in a radioactive area. This result provides essential information in describing radioactive aerosols in Fukushima.

## 1 Introduction

Aerosols are solid and liquid particles suspended in air and are composed of organic compounds, inorganic salts, trace elements, black carbon, water, and other substances (Reggente et al., 2019). Aerosols can be transported from their sources, which include natural (e.g., volcanos, dust, ocean) and anthropogenic emissions, to different locations and are susceptible to environmental

and weather conditions. Other studies have shown that aerosols and their optical properties in a particular area show seasonality (Humphries et al., 2023; Orikasa et al., 2020; Jain et al., 2020; Cigánková et al., 2021).

In some cases, in radioactive areas, radioactive aerosols and dust that come from the ground are also transported. Radioactive aerosols are particulates that have been exposed to nuclear radiation and thus have become radioactive. In a place where the unprecedented release of nuclear radiation brings about radioactive aerosols, these can pose a risk to nearby inhabitants. Past studies have evaluated radiation using gamma-ray detectors and measured radiation dose rates at different sites to analyze radiation values from dust and infer radioactive dust's wind transport (Allott et al., 1992; Takagi et al., 2019; Yamauchi, 2012). These techniques are adequate, especially when evaluating the safety of an area exposed to radiation for people to return (Matsuo et al., 2019). However, since the atmosphere is dynamic, wind transports these radioactive aerosols to different places. Monitoring these transport in a higher temporal scale can be an essential observation to assess radioactivity in an area. Remote sensing instruments, like lidars, that can perform continuous and automated measurements, gathering data with minimal human interference are valuable in monitoring radioactive aerosols. In this work, the authors will show that the horizontal lidar can be used for this kind of monitoring.

Lidar networks are used to monitor vertical distributions of dust and aerosols in the atmosphere (Campbell et al., 2002; Kawai et al., 2018; Pappalardo et al., 2014) and validate satellite (Kim et al., 2008; Whiteman et al., 1990; Gusmão et al., 2020). But lidar systems, together with other in situ instruments, can be deployed in radioactive areas and operated continuously with less human intervention and operation. Such applications are the main focus of this paper. Theoretically, continuous detection of these radioactive aerosols and dust using lidar can reveal valuable information in understanding and inferring temporal or diurnal evolution, the effect of seasonal changes, and the relationship between meteorological and optical parameters on observed naturally occurring local aerosols and dust. Lidar can also provide information on the transport of radioactive aerosols. In Chiba, the seasonality of aerosols has been reported (Fukagawa et al., 2006), while in Fukushima, the seasonality of aerosols is still to be quantified.

After the radiation incident in Fukushima area in 2011, places near the radioactive meltdown have become uninhabited (Hidaka et al., 2022). However, years after the incident, radioactivity has declined. Still, many of the inhabitants have a hard time returning due to various reasons such as absences of family members and neighbors, and insufficient radiation decontamination, among others (Matsuo et al., 2019). In our previous study, the horizontal lidar was operated near an interim storage facility for a few days in Fukushima, Japan, in 2018 to monitor the optical scattering of dust during the storage operation or working conditions (Shiina et al., 2018). The study showed that dust smaller than 20 $\mu$m in diameter carries more radioactivity ($\sim 2 \times 10^{-8}$ Bqcm$^{-3}$) than larger dust (Shiina et al., 2018). This amount is much lower than the accepted safe limit of 10 $\times 10^{-5}$ Bqcm$^{-3}$. However, continuous observations of aerosols in radioactive areas are still essential to determine the amount of daily inhalable radioactive dust, to reassure residents that radioactivity levels from inhalable aerosols are safe, and to promote safe everyday life activities in these communities. Previous studies indicate that inhaled radioactive aerosols under a dust concentration of $1 \times 10^{-4}$ gm$^{-3}$ can have an effective dose of just 5.6 $\mu$Sv received in 20 years by an adult with light activity. This dosage is smaller than the total effective dose of 9.9 $\mu$Sv and 48.8 $\mu$Sv for adults with moderate and vigorous activities, respectively, during the same number of years at an air dust concentration of $1.5 \times 10^{-4}$ gm$^{-3}$ (Hanfi et al., 2021; Valentin,

2002). As people gradually return to evacuated places in Fukushima, continuous monitoring of natural radioactive aerosols assists in assessing these areas' environmental and living conditions. Discussion of monitoring radioactive aerosols and dust, the phenomenological aspect of monitoring, and data acquisition and interpretation provide essential insights into the current and future detection of radioactive aerosols. Furthermore, having a dataset of optical properties of local aerosols and quantified aerosol-weather relationships helps understand the local transport of these aerosols, aids in analyzing radiative forcing, air quality, health impacts on the residents, precipitation patterns, atmospheric dynamics, validating other remote sensing data, and can be used for validating and improving local climate models. Combining these with the data from ancillary instruments data from weather monitors, aerosol samplers, and chemical analysis, monitoring the flow of radioactive aerosols can be analyzed to infer possible locations of hot spots.

The main objective of this work is to provide an overview of the results obtained during the demonstration experiment phase of the project. This work has the following objectives: 1) to present continuous data from a near-ground horizontally pointing lidar system, 2) to analyze temporal changes in the optical properties of local aerosols, and 3) to quantify the relationship between the observed aerosol optical properties and other meteorological parameters (e.g., relative humidity (RH), wind speed, particulate matter (PM$_{2.5}$).

## 2 Site description

The lidar is operated on the 9th floor ($\sim$ 65 m above sea level and 54 m above the ground) of the Engineering building at Chiba University (35.6278° N, 140.1031° E). The observation site is located in an urban area 3 km from Tokyo Bay. Within a 10 km radius, the north and east sides are urban areas. In contrast, the west and south areas comprise urban lands and Tokyo Bay (Fig. 1). The landmass and water locations make the observation site a strategic place for measuring the optical properties of aerosols coming from different directions.

The planned observation site will be in Fukushima, northeast of and $\sim$ 165 km from Chiba. The specific site location will be decided in the future, but the site will be in between uninhabited and inhabited areas. In Fukushima, the east side of the site is looking at the Pacific Ocean. The north and west sides are surrounded by landmasses. The south side is covered by both land and sea. The aerosol optical properties derived from the data from long-term monitoring of aerosols in this area are expected to be affected by winds coming from the sea and the land, similar to the case in Chiba.

## 3 Methodology

In this work, a horizontal lidar, a weather monitor, and a particulate matter (PM$_{2.5}$) sampler are simultaneously and continuously operated. The data from these instruments are processed and compared to infer the relationship between lidar-derived aerosol optical and meteorological parameters. The robust fitting method is employed to infer quantitative relationship trends while ignoring outlier points to quantify the effect of weather on the optical properties of near-ground aerosols. The variation of each optical and weather parameter can be considerable due to the high variability of atmospheric conditions, but trends

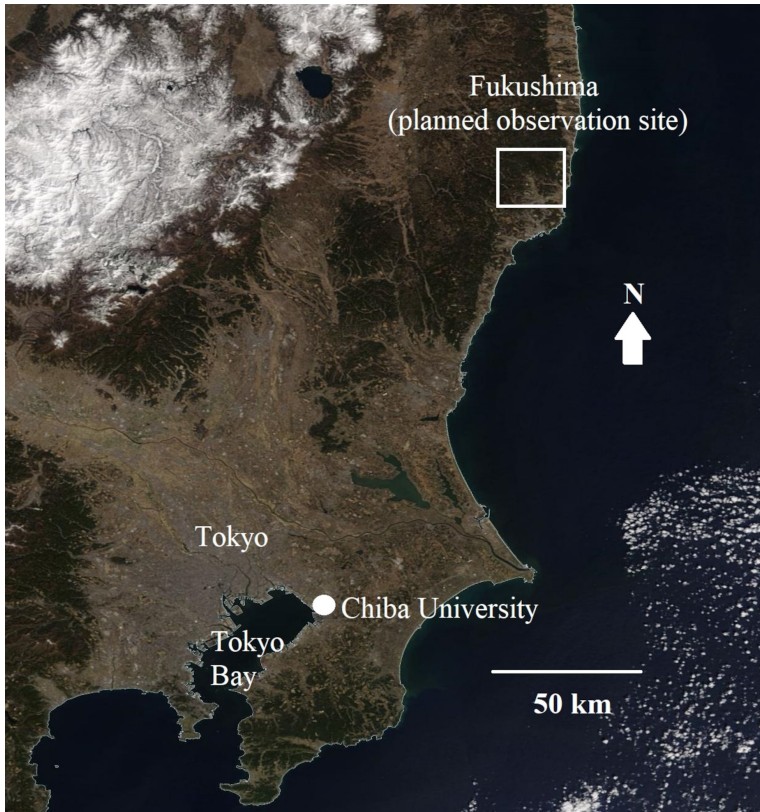

**Figure 1.** Location of Chiba University, the current observation site, with respect to Tokyo and Tokyo Bay. The next planned observation site will be in Fukushima (https://worldview.earthdata.nasa.gov/).

can be observed. In some cases, using robust fitting can result in computational complexity and efficiency loss. However, these were not encountered when analyzing the data.

## 3.1 Horizontal lidar

The horizontal lidar has been operated continuously from August 2021 to February 2022 on the 9th floor of the Engineering building at Chiba University (35.63$^o$N, 140.1$^o$ E) to monitor near-ground aerosols and dust. The lidar system gathers data every 1 s and is only stopped for a few minutes when the lidar is reset once every other day. The lidar is set in a perfectly horizontal position and is pointing in the west-northwest direction. In Fukushima, the lidar will be mounted closer to the ground ($\sim$ 1 m). Table 1 shows the specifications of the horizontal lidar system. A diode-pumped solid-state (DPSS) laser is used as a light source, and it emits a maximum pulse energy of 120 $\mu$J at a repetition frequency of 1 kHz at 349 nm, with a full beam divergence of 3 mrad prior to expansion. The transmitted beam is expanded to 30 mm in diameter and is considered not safe at near distance. The laser and telescope's full overlap is around 50m from the lidar. The transmitted and received signals are

P-polarized and P and S-polarized, respectively. The lidar data are then processed for extinction coefficient and depolarization ratio.

**Table 1.** Specification of the DPSS horizontal lidar system

| | |
|---|---|
| Laser model | Spectra-Physics Explorer One |
| Wavelength | 349 nm |
| Pulse energy | 120 $\mu$J |
| Repetition frequency | 1 kHz |
| Pulse duration | <5 ns |
| Beam divergence | 3 mrad |
| Telescope diameter | 10 cm |
| Telescope's field of view | 3 mrad |
| Interference filter CWL | 349 nm |
| Interference filter FWHM | 0.6 nm |
| PMT Hamamatsu | H11901-110 |
| Transmitted polarization | P |
| Received polarization | P and S |
| Maximum range | 700 m |

Since the horizontal lidar has a maximum range of around 700 m, we assume that the aerosol properties do not significantly vary in a few hundred meters in the horizontal direction. From this assumption, the aerosol extinction coefficient between 100 m and 300 m can be derived using the slope method (Kunz and de Leeuw, 1993). The slope method is used by fitting a line between the maximum and minimum of the natural logarithm of the range-corrected signal between 100 m and 300 m. A homogeneous atmosphere can be assumed during the observation period in this horizontal range. This method is similar to our group's work that examined the relationship between horizontal lidar and weather effects from continuous lidar observations in November 2017 and in the winter of 2018 (Ong et al., 2019; Xiafukaiti et al., 2020).

The volume linear depolarization ratio (VLDR) from the lidar signals is calculated as the ratio of the S and P signals (Freudenthaler et al., 2009; Comerón et al., 2018) multiplied by the ratio of the gains of the channels. Burton et al. (2015) and Xu et al. (2022) presented a method to measure the ratio of the gains of the channels, and this method is applied in this work. The gain is obtained by measuring the signals from each channel when the polarizer is oriented at the parallel and perpendicular positions. Table 2 shows the days in a month when the horizontal lidar continuously gathers data.

## 3.2 Weather data

A weather monitor (Davis, Vantage Pro 2) is mounted on the rooftop ($\sim$ 46 m above sea level) of the Center for Environmental Remote Sensing (CEReS). The weather monitor is installed at around 74 m horizontally and 19 m from the lidar system. It logs data on weather parameters (RH, temperature, wind speed, wind direction, and rain rate) every 5 min. The absolute humidity

**Table 2.** Number of days of continuous operation of horizontal lidar

| Month | Number of days (%) |
| --- | --- |
| August 2021 | 20 (64.5) |
| September 2021 | 30 (100) |
| October 2021 | 20 (54.5) |
| November 2021 | 27 (90) |
| December 2021 | 28 (90.3) |
| January 2022 | 29 (93.6) |
| February 2022 | 23 (82.14) |

can be derived from the RH and temperature data (Seinfeld and Pandis, 2016). This parameter, which provides the amount of water vapor per unit volume of air, can be compared with the optical parameters from the lidar.

### 3.3 PM$_{2.5}$ data

PM$_{2.5}$ data are obtained using a compact PM$_{2.5}$ instrument (Nakayama et al., 2018; Damiani et al., 2021). This instrument is programmed to get PM$_{2.5}$ data every 1 min on the 9th floor of the Engineering building at Chiba University as part of the international observation network activities of the Sky Radiometer Network (SKYNET) (Nakajima et al., 2020; Hashimoto et al., 2012) and Aerosol and Sky Research Remote Sensing Network (A-SKY). The SKYNET network uses the PREDE POM-01 and POM-02 as in-situ sky radiometers.The PM$_{2.5}$ instrument does not show hygroscopic effects for RH<70%.

### 4 Results and discussion

Weather conditions affect aerosols' optical and radiative properties (Makar et al., 2015, 2021; Mao et al., 2018). In this work, the effect of weather on aerosol optical properties can be quantified using the data from the horizontal lidar and weather monitors. The following narration shows the temporal changes and the quantified relationships between lidar and weather monitors obtained by applying robust fitting on the parameters.

### 4.1 Observed temporal trends

During the seven months of continuous operation of the horizontal lidar, average extinction coefficients and depolarization ratio show decreasing and increasing monthly trends, respectively (Figs. 2a and 2b). These results indicate the optical parameters' dependence on seasons. This seasonal dependence was also observed in previous studies using a chemical sampler, chemical analysis, weather data, and sun photometer (Fukagawa et al., 2006). In this work, the temporal trend of the mean extinction coefficient near the ground slightly decreases during cold months (November to February). The distribution of extinction coefficients during the cold months also tends to be skewed towards higher values due to high outlier values. For all

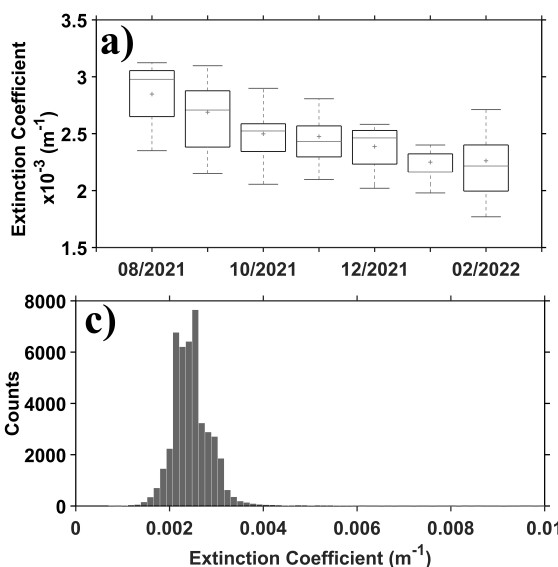
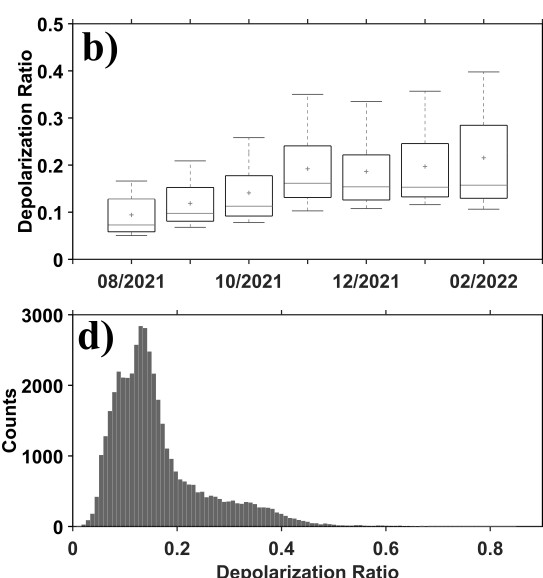

**Figure 2.** Boxplots of monthly a) extinction coefficient, and b) depolarization ratio derived from horizontal lidar in the 100 m to 300 m range and the corresponding histogram of the c) extinction coefficient and d) depolarization ratio.

collected datasets, the observed median and average extinction coefficients of aerosols within the 100m to 300 m range are
below 0.005 m$^{-1}$. The mean extinction coefficient value is around 0.0025 ($\pm$ 0.00048) m$^{-1}$. The histogram of the extinction coefficient values is shown in Fig. 2c and suggests a two-mode distribution. A simple bimodal Gaussian fit of the distribution (not shown) indicates mean extinction coefficients of 0.0024 and 0.0027 m$^{-1}$. These very close mean extinction coefficients can be attributed to the effect of relative humidity (RH), as discussed in the succeeding paragraph. The extinction coefficient values presented in this work are roughly ten times greater than previously reported works (Ong et al., 2019; Xiafukaiti et al.,
2020). In these previous works, a laser inclined 4° above the horizontal was used to observe near-horizontal aerosols. In this configuration, the laser traverses the atmosphere with a vertical path, encountering fewer aerosols along its path. The extinction coefficient is measured up to 5 km by accumulating the signals over 5 min and continuously operated in the months of November 2017 (Ong et al., 2019) and November and December 2018 (Xiafukaiti et al., 2020). The current horizontal lidar does not have this property. The number of aerosol particles does not significantly vary between the 100 m to 300 m horizontal
range, leading to higher extinction coefficient values.

The mean depolarization ratio values increase from summer to winter (Fig. 2b). The mean depolarization ratio is 0.161 ($\pm$0.095). The histogram of the depolarization values is shown in Fig. 2d. Between the depolarization range of 0 and 0.2, a bimodal distribution can can be observed. The result of applying a bimodal Gaussian fitting of the distribution reveals a mean depolarization of 0.12 and 0.14. These mean values approximate the local depolarization ratio values during high and
low humidity conditions. The mean depolarization ratio between 0.35 and 0.4 can be attributed to highly non-spherical dust. The mean depolarization values between 0.1 and 0.35 are similar to the results obtained from dual-polarization (volume linear

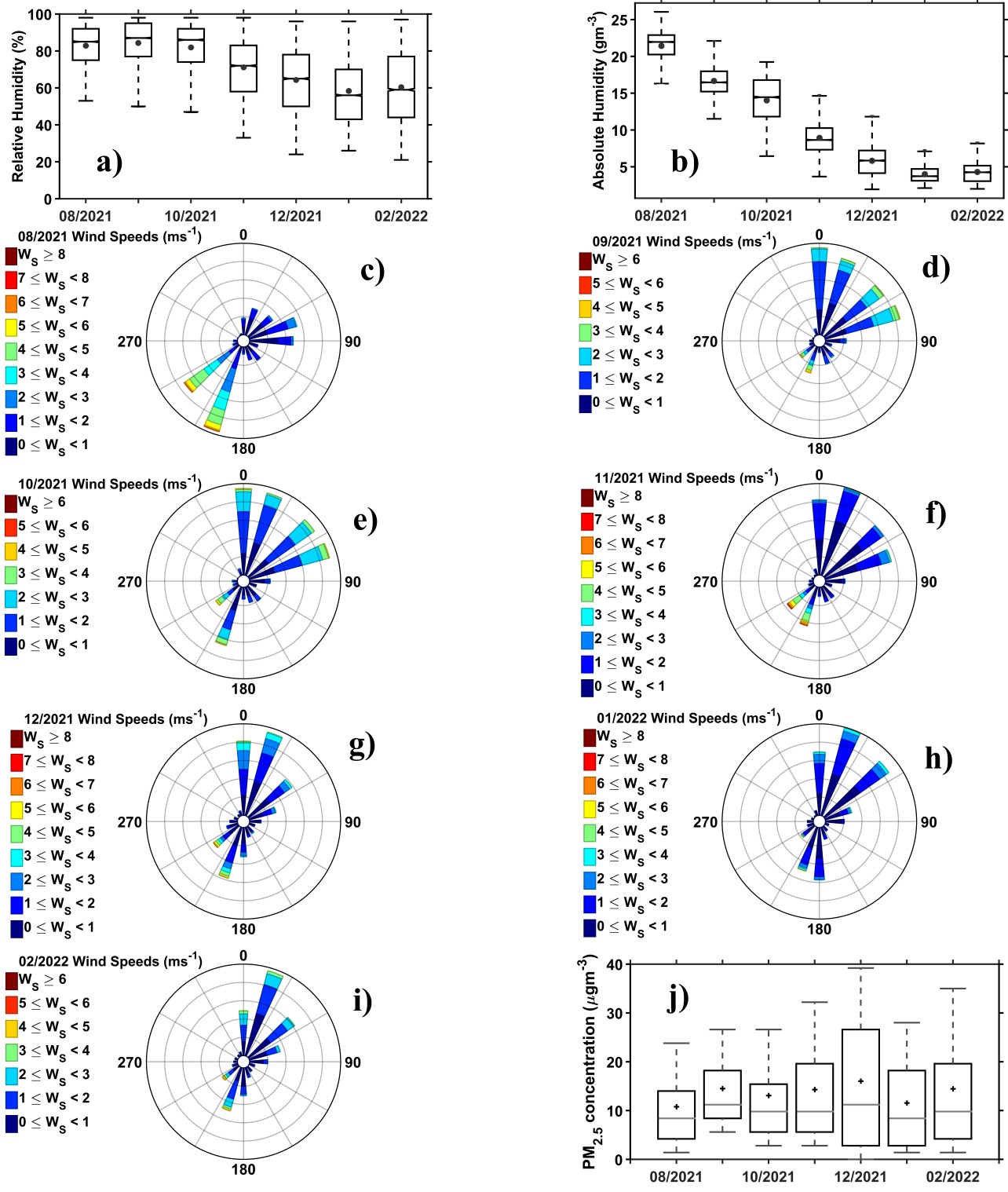

**Figure 3.** Monthly distribution of a) RH, and b) absolute humidity, and rose graph of of wind speed and direction for c) August, d) September, e) October, f) November, g) December 2021, and h) January and i) February 2022. Monthly distribution of j) average PM$_{2.5}$ concentration shows values less than 20 $\mu$gm$^{-3}$.

depolarization ratio) lidar measurements (Qi et al., 2021) and from laboratory measurements (linear depolarization ratio) of mineral dust fitted with a homogeneous spheroid model (Kahnert et al., 2020). In the latter, an increase depolarization ratio can be attributed to the increase in the aspect ratio of the dust for a constant size parameter. Previous works on dust observations have reported particle linear and volume depolarization ratios from 0.2 to 0.4 at 355 nm (Haarig et al., 2022; Huang et al., 2023). In real conditions, this effect can occur when RH is low and the prevailing aspect ratio of dust is not equal to unity.

From the weather monitor data, continuous measurements of relative humidity (RH) from August to February show a decrease in average RH from 80% to 60% (Fig. 3a). This corresponds to the reduction of absolute humidity from 22 $\mathrm{gm^{-3}}$ to 4 $\mathrm{gm^{-3}}$ in the same period (Fig. 3b). In August, the dominant wind direction is from the south and southwest (Tokyo Bay direction), carrying more moisture and marine-type aerosols (Fig. 3c). In the subsequent months, the prevailing wind directions drastically shift from southwest to northeast and north (Figs 3d-3i). Winds from the north come from the land, and the aerosols can be considered an urban type (Fukagawa et al., 2006). The observed mean $PM_{2.5}$ concentration is below 20 $\mu\mathrm{gm^{-3}}$ (Fig 3j). It is worth mentioning here that during the observation period, no nearby volcanic eruption occurred during the observation period. Pollen usually occurs during springtime. Thus, to the authors' knowledge, the sources of non-spherical aerosols during the observation period are from dust, sea salt and snow.

The wind direction plays a critical role in interpreting the aerosol data from areas affected by radioactivity. In Chiba, the dominant winds from the south and southwest indicate the presence of marine-type aerosols. On the other hand, when the wind is blowing from the east to the west in Fukushima, it is anticipated that the observed aerosols will primarily consist of marine-type aerosols, assuming that radioactive aerosols originate from land-based sources. Properly evaluating the impact of wind direction on aerosol presence and distribution is crucial for obtaining accurate insights into the atmospheric conditions in the area and interpreting the aerosol data effectively.

## 4.2    Relationship between lidar-derived optical parameters (depolarization ratio and extinction coefficient), weather parameters (RH, absolute humidity, wind speed, wind direction), and $PM_{2.5}$ concentration

The overall data from ground observations of near-ground aerosols indicate that the depolarization ratio decreases exponentially as the extinction coefficient increases, as shown in the orange circles in Fig. 4a. This relationship between the two optical parameters indicates that aerosols grow in size as a response to increasing RH. As the aerosols grow in size, the sphericity is also affected since water uptake results to more spherical particles as a result of water covering or diluting the condensation nucleus (Tang et al., 1997, 2019; Wang et al., 2022). The high depolarization ratio (>0.3) with extinction coefficients from 0.003 to 0.006 $\mathrm{m^{-1}}$ (black circles in Fig. 4a) is primarily from the days when snowflakes fell and when wind speed was high (> 0.5 m), bringing in local dust. Snow was observed in Chiba on 06 January 2022 (15:00-00:00 JST), 07 January 2022 (01:00-13:00 JST), 02-03 February 2022 (22:00-11:00 JST), and 14 February 2022 (00:00-07:00). Just before snow conditions, the observed data suggests that the depolarization ratio decreases exponentially from 0.4 to 0.1 when the extinction coefficient changes from 0.002 $\mathrm{m^{-1}}$ to 0.003 $\mathrm{m^{-1}}$. When the extinction coefficient is greater than 0.003 $\mathrm{m^{-1}}$, the depolarization ratio varies from 0.002 to 0.8, suggesting the formation of different snowflake shapes. Changes in the depolarization ratio for aerosols tend to follow the exponential decrease with the extinction coefficient. Dust is also found to follow the exponential decrease with

extinction coefficient (black points coexisting with the orange points when the extinction coefficient is less than 0.003 m$^{-1}$). The high depolarization ratio and low extinction coefficient can be attributed to the contribution of marine-type aerosols, as will be shown later when these optical parameters are compared with wind direction. The quantified relationship between the depolarization ratio and extinction coefficient serves two purposes. First, the quantified results provide, for the first time, a measured relationship between the two parameters, local and near-ground aerosols. Second, the results provide an essential understanding of how radioactive near-ground aerosols can be observed as they change shape due to the effect of weather. Previous results have made indirect comparisons from vertically-pointing lidars, i.e., no quantitative relationship is provided (Groß et al., 2011; Haarig et al., 2017, 2022; Sakai et al., 2006).

When the extinction coefficient is compared with RH (Fig. 4b), an exponential function can be used to construct the relationship between the two parameters for aerosols and dust near the ground. This result is expected because when RH increases, the optical properties of the scatterers change, making these scatterers have a greater extinction cross-section that causes high attenuation of light (Tang et al., 1997, 2019). Other studies have shown that the extinction coefficient and RH relationship can be modeled using a power law applied to RH/100 or (1-RH/100) (Skupin et al., 2016; Düsing et al., 2021). According to these studies, a high extinction coefficient at high RH does not mean high wind speed but depends on the aerosol type. For example, at 90% RH, the extinction coefficient and growth rate of marine-type aerosols are higher than dust (Zieger et al., 2013). However, for near-ground aerosols, the exponential function better fits the seven-month observation.

The extinction enhancement factor, f(RH), defined as the ratio of the extinction coefficient at a specific RH to the extinction coefficient at dry RH, is related to the hygroscopic growth of aerosols under increasing RH (Haarig et al., 2017; Skupin et al., 2016; Zhang et al., 2022). Previous studies use only the scattering coefficient of aerosols to determine f(RH) to characterize the hygroscopic properties of aerosols (Burgos et al., 2019; Jung et al., 2021; Lagrosas et al., 2019; Zieger et al., 2011, 2013). The extinction coefficients provide the trend of the effect of RH on near-ground aerosols. In this work, the f(RH) values (Fig. 4c), measured at 349 nm, range from 1 to 2 and do not show a steep exponential increase with RH. This range of values is smaller than the f(RH) values at 550 nm (Jung et al., 2021; Liu et al., 2023; Zieger et al., 2014) and is consistent with the f(RH) values at 355 nm reported by Haarig et al. (2017). Also, earlier measurements of dry and wet marine aerosols have f(RH) of 1.94 ± 0.94 between 40% and 80% RH at 355 nm (Haarig et al., 2017). In this case, for pure aerosol types, such as the case of marine aerosols, f(RH) is higher than the values shown in Fig. 4c in the same RH range. The result presented in Fig. 4c represents Chiba's unique ambient near-surface f(RH) from different aerosol types.

The mean depolarization ratio of near-surface aerosols shows a monotonic decrease as RH increases (Fig. 4d). The results suggest that water vapor in the atmosphere can control, to some extent, the shapes of near-ground aerosols and dust by making them approach a spherical shape as RH increases. An exponential function is found to provide the best fit and thus model the relationship between the depolarization ratio and RH. The variation of the depolarization ratio when RH < 60% is higher than the variation when RH > 60%. This observation implies that various aerosols with different non-spherical morphologies in dry conditions (RH < 60%) are observed. As deliquescence sets in, condensing water vapor molecules change the morphological structure of the aerosols. Around RH = 100%, the mean depolarization ratio is around 0.1. This result is expected since the horizontal lidar observes near-ground aerosols. Furthermore, the 0.1 average depolarization ratio indicates the limiting value

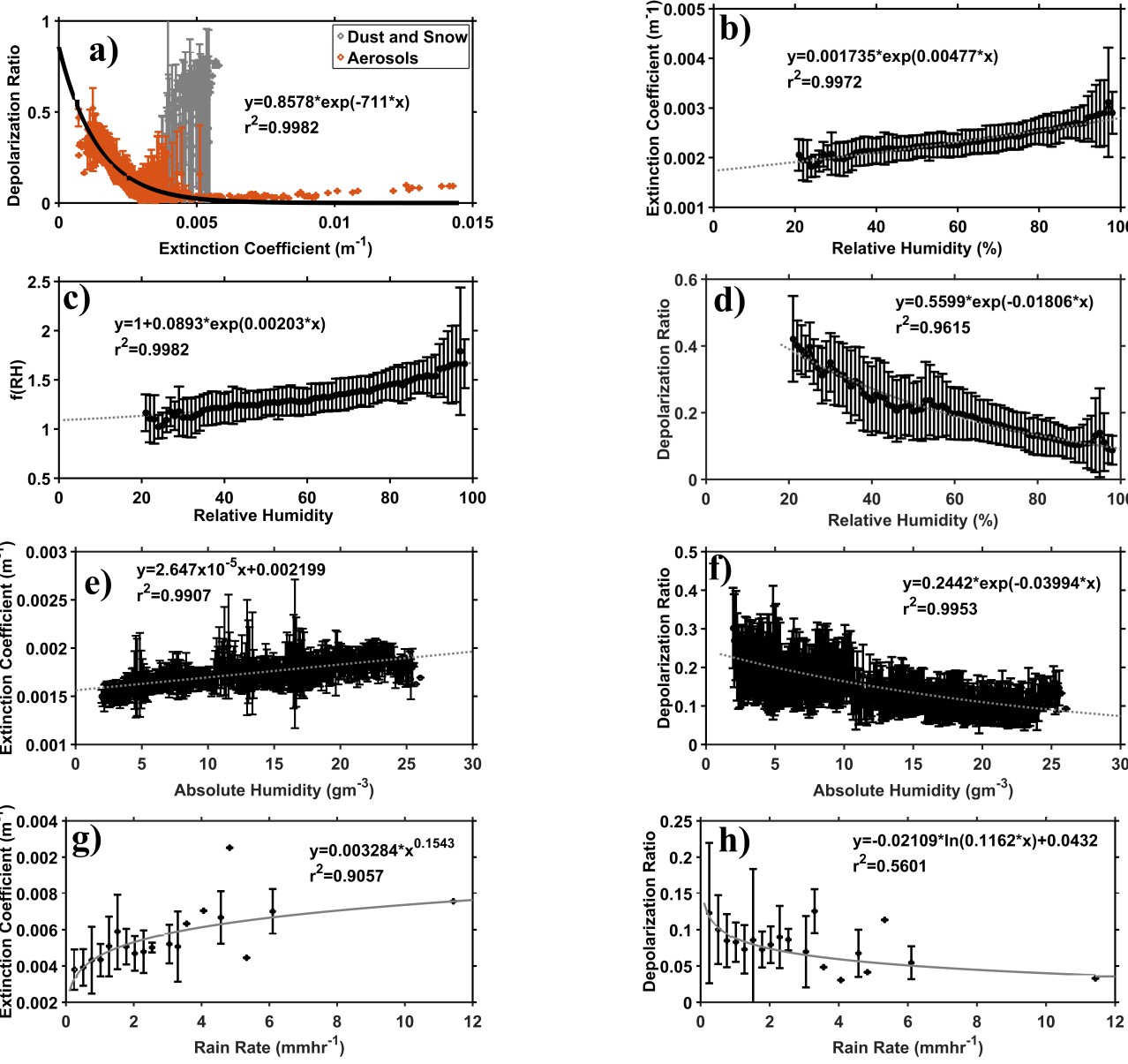

**Figure 4.** Observed relationship between a) depolarization ratio and extinction coefficient, and b) extinction coefficient, c) f(RH) and d) depolarization ratio with RH, e) extinction coefficient and f) depolarization with absolute humidity in Chiba. During rainy conditions, g) extinction coefficient, and h) depolarization ratio can be quantified with rain rate.

for near-ground aerosols and implies that they cannot possess a perfectly spherical shape even when coated by water at high RH. This result provides new insight into how radioactive aerosols can be affected by RH, especially when the lidar system is placed near the ground, as will be the case in Fukushima.

The quantified relationship between extinction coefficient and depolarization ratio with RH can be further explored by link-
ing these lidar-derived optical parameters with the amount of water vapor in the atmosphere by employing absolute humidity. Absolute humidity is the parameter that measures the actual amount of water vapor regardless of the air's temperature. A linear relationship between the extinction coefficient and absolute humidity is expected since the extinction coefficient can be linked with the aerosol's mass using the mass extinction efficiency parameter (Cheng et al., 2017; Latimer and Martin, 2019; Lagrosas et al., 2005). The quantified relationship between extinction coefficient and absolute humidity from lidar and weather monitor
measurements is the first to be reported from the authors' knowledge. Previous works usually employ the use of the hygro- scopic growth factor, f(RH), to describe and quantify the change of extinction coefficient with relative humidity (Chen et al., 2019; Dawson et al., 2020; Jefferson et al., 2017; Lagrosas et al., 2019; Ming and Russell, 2001; Noh et al., 2011; Shingler et al., 2016; Vu et al., 2021; Zieger et al., 2011, 2013). The seven-month measurements have shown that extinction coefficients increase from 0.002 $m^{-1}$ to 0.004 $m^{-1}$ when absolute humidity increases from 3 $gm^{-3}$ to 25 $gm^{-3}$ (Fig. 4e). The aerosol
extinction coefficient, therefore, increases by 2.647 $\times 10^{-5} m^{-1}$ per 1 gm-3 increase of water vapor. On the other hand, the change of depolarization ratio with absolute humidity can be modeled as an exponential function, indicating that near-ground aerosols are sensitive to condensing water vapor (Fig. 4f). These relationships provide insights into the amount of water vapor that can effectively change the optical characteristics of aerosols. Previously reported classification of aerosols in Chiba by air sampling has shown that hydrophilic aerosols (sea salt, ammonium nitrate, and ammonium sulfate) are dominant during
summer months, while elemental carbon is the dominant aerosol during winter months (Fukagawa et al., 2006). These aerosols can potentially have strong and weak interactions with water vapor in the atmosphere.

During rainy conditions, the extinction coefficient values derived from lidar data vary from 0.003 $m^{-1}$ to 0.005 $m^{-1}$ as rain rate increases (Fig. 4g). Depolarization ratio values decrease from 0.12 to 0.003, as expected, with increasing rain rate (Fig. 4h). These extinction coefficients and depolarization ratio values also coincide with the same set of points at the bottom portion
of Fig. 4a just before the depolarization ratio goes variably high (black points). This result implies that these values can be considered the threshold values before the water freezes and forms snowflakes.

$PM_{2.5}$ concentration measured during the seven-month observation shows increasing concentration as RH increases (Fig. 5a). At RH>80%, the mean $PM_{2.5}$ concentration deviates from the trend. This observation can be attributed to the fact that the $PM_{2.5}$ values for high RH (>70%) are affected by hygroscopic growth (Nakayama et al., 2018). Previous works have
observed this deviation at RH=70% (Lou et al., 2017; Wang and Ogawa, 2015; Zalakeviciute et al., 2018) and have attributed this to the effects of rain, hygroscopic growth that leads to reducing $PM_{2.5}$ concentrations. Even though, this result implies that in an area where radioactive aerosols exist, the increase in $PM_{2.5}$ concentration can be due to the increase in RH and not necessarily an increase in radioactive aerosols unless the water vapor is also radioactive or when the prevailing wind comes from the land which is the source of radioactive aerosols and dust. When the horizontal lidar continuously observes aerosols
nearer the ground, as will be the case in Fukushima, a similar relationship between $PM_{2.5}$ and RH is expected. It is worth

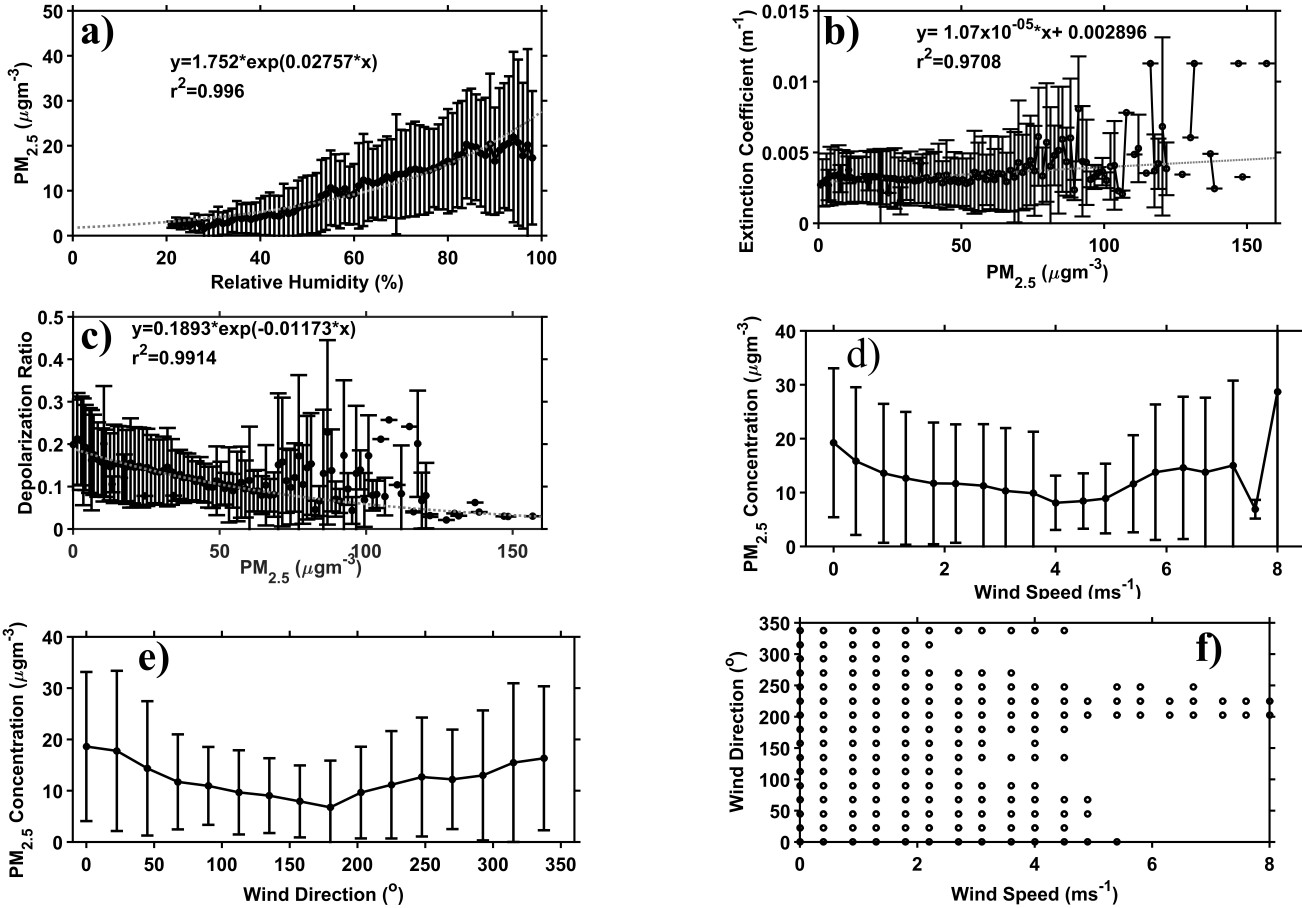

**Figure 5.** a) Observed relationship between PM$_{2.5}$ and RH and b) extinction coefficient and depolarization ratio with PM$_{2.5}$ and the corresponding trend of PM$_{2.5}$ with d) wind speed and e) wind direction and f) wind direction with wind speed.

mentioning that the fitted equations in the figures are only valid in the measured range of values, e.g., 20 $\mu$gm$^{-3}$ < PM$_{2.5}$ <
160 $\mu$gm$^{-3}$. When the extinction coefficient is compared to the PM$_{2.5}$ concentration (Fig. 5b), a linear fit can be anticipated
since the extinction coefficient is also proportional to the exponent of RH and the PM$_{2.5}$ concentration is linearly proportional
to the extinction coefficient. The extinction coefficient from previous studies shows a linear relationship with PM$_{2.5}$ with a zero
offset. This result is expected because corrections from contributions from nitrogen dioxide on PM$_{2.5}$, and PM$_{10}$ concentrations
are applied (Cheng et al., 2017; Gramsch et al., 2004; Yao et al., 2021). On the other hand, without any corrections applied to
the data, other results show a similar non-zero intercept relationship between PM$_{2.5}$ and aerosol optical parameters (aerosol
optical depth, absorption, and extinction coefficients ) (Kong et al., 2017; Mei et al., 2017; Yang et al., 2019). In this work, no
corrections are implemented since this study focuses on the atmospheric effects on ambient aerosols and dust. This quantified
relationship enables the approximate conversion of measured PM$_{2.5}$ and extinction coefficient.

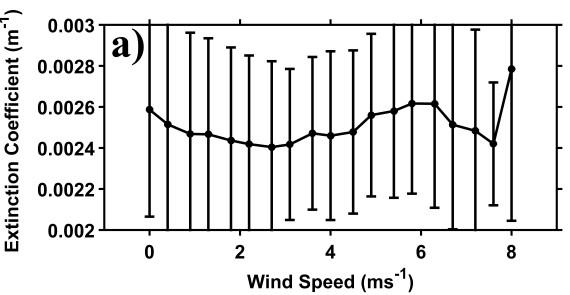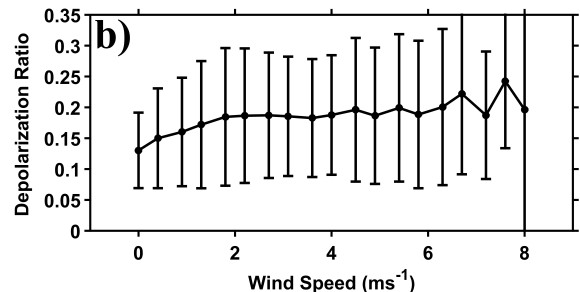

**Figure 6.** Variation of average a) extinction coefficient and b) depolarization ratio with wind speed.

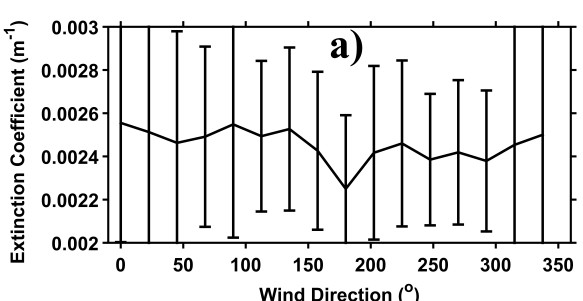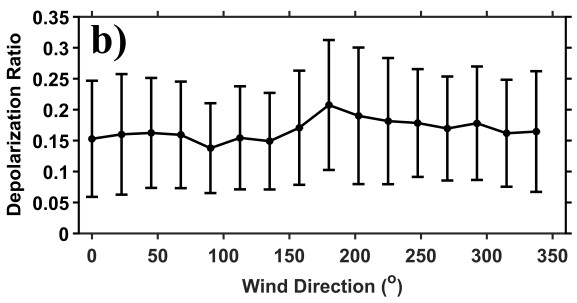

**Figure 7.** Variation of average a) extinction coefficient and b) depolarization ratio with wind direction.

The depolarization ratio from observed local aerosols decreases with increasing $PM_{2.5}$ and can be modeled using an exponential function (Fig. 5c). Increased $PM_{2.5}$ concentrations indicate aerosol growth by deliquescence, i.e., formation of more near-spherically shaped aerosols with sizes less than 2.5 $\mu$m. However, higher depolarization values are observed for $PM_{2.5}$ concentrations greater than 60 $\mu$gm$^{-3}$ deviating from the observed trend. These conditions are generally detected when wind

speeds are high ($> 0.5$ms$^{-1}$). This result is also observed in the extinction coefficient and $PM_{2.5}$ graph (Fig. 5b). Wind speed greater than 0.5 ms$^{-1}$ are associated with different $PM_{2.5}$ concentrations. The observations in Chiba show that the distribution of $PM_{2.5}$ with wind speed has a V-like structure with the wind speed 4 ms$^{-1}$ serving as the inflection point and further indicates a bimodal distribution (Fig. 5d). $PM_{2.5}$ values peak in the range from 1 ms$^{-1}$ to 2 ms$^{-1}$ and from 5.5 ms$^{-1}$ to 7.5 ms$^{-1}$. This information shows that the wind easily carries small dust near the ground and is dispersed as the wind speed increases. The

second peak indicates increased $PM_{2.5}$ concentrations from local or long transport. $PM_{2.5}$ concentrations also have a bimodal distribution with wind direction (Fig. 5e). $PM_{2.5}$ concentrations peak when the wind comes from the northeast, the location of a major highway, and the south-southwest, the location of Tokyo bay and partly urban land. Most wind speeds greater than 5 ms$^{-1}$ also come from the south-southwest direction (Fig. 5f). The increase in $PM_{2.5}$ concentration at higher wind speeds implies the abundance of local or coated aerosols from the water vapor coming from Tokyo bay. The seven-month observation

shows land-based $PM_{2.5}$ concentrations are higher by $\sim 50$ $\mu$gm$^{-3}$ than Tokyo bay-based aerosols.

When average extinction coefficients and depolarization ratios are compared with wind speed, patterns of the average values are noticeable even though high variations exist (Figs 6a and 6b). These high variations are expected when long-term atmospheric data are analyzed. The average extinction coefficient tends to have a bimodal distribution with wind speed similar to that of the $PM_{2.5}$ concentration with wind speed. Average extinction coefficients are observed to peak when wind speed is less

than $0.5$ ms$^{-1}$ and between $5$ ms$^{-1}$ and $7$ ms$^{-1}$ (Fig. 6a). The first peak can be attributed to local aerosols near the lidar system while the second peak can be attributed to both marine-type aerosols and dust. Wind speed greater than $5$ ms$^{-1}$ indicates wind directions from $200^o$ to $250^o$. This result is anticipated since the extinction coefficient is shown to have a linear relationship with $PM_{2.5}$. Similarly, the values of the average depolarization ratio increase with wind speed (Fig. 6b). The relationship between the average depolarization ratio and wind speed demonstrates that the wind effectively carries more nonspherical dust

in the atmosphere, and in Chiba, a wind speed increase of $1$ ms$^{-1}$ increases the average depolarization ratio by approximately $0.008$. This result is consistent with the previous discussion wherein an increase in the depolarization ratio indicates a decrease in $PM_{2.5}$ is ascertained and a reduction in $PM_{2.5}$ concentration suggests a more likely increase in wind speed.

When the average extinction coefficient is compared with wind direction, a slight decrease of the average extinction coefficient with wind direction can be observed from $0°$ to $315°$ (Fig. 7a). This decrease is approximately $-3.3 \times 10^{-7}$m$^{-1}$ per 1°.

Conversely, the average depolarization ratio moderately increases with wind direction at a rate of $6.8 \times 10^{-5}$ per 1° (Fig. 7b). Between $0°$ and $150°$ both parameters show slight changes in values. The average extinction coefficient decreases a bit from $2.4 \times 10^{-3}$ m$^{-1}$ to $2.2 \times 10^{-3}$ m$^{-1}$ when the wind direction is from $180°$ to $225°$ i.e., from Tokyo bay. The average depolarization ratio increases by $0.05$ in the same wind direction range, and this high average depolarization ratio ($\sim 0.2$) implies the dominance of marine-type aerosols in the atmosphere. The corresponding decrease in the average extinction coefficient

when the wind is from Tokyo Bay suggests that marine-type aerosols have a lower extinction coefficient value ($\sim 0.0023$ m$^{-1}$) than land-based aerosols. Wind direction between $225°$ and $315°$ carry aerosols with lower extinction coefficients than aerosols from $0°$ to $150°$. Conversely, the average depolarization ratio shows a relatively higher value ($\sim 0.2$) when the wind direction is between $225°$ and $315°$ than aerosols from $0°$ to $150°$ and can probably be attributed to the higher wind speeds between $200°$ and $250°$ that can be responsible for transporting more aerosols either from local or farther places.

These overall results provide information and insights into how the lidar-derived optical properties of aerosols are affected by atmospheric parameters (RH, absolute humidity, wind speed, wind direction) and $PM_{2.5}$. The quantified relationships can aid and provide a general understanding of Fukushima's aerosol-weather relationship. When operated in Fukushima, the horizontal lidar can observe aerosols from the land and sea, and we expect that lidar-weather parameters can also be quantified. As mentioned in Sec. 2, the left and right sides of the deployment site are land and the Pacific Ocean, respectively. Sources

of radioactive aerosols are the aerosols from the land. Wind direction and speed will be the essential weather parameters that can be used to determine whether the observed aerosols come from the land or the Pacific Ocean. The observations in Chiba show that aerosols from Tokyo Bay have lower extinction coefficients and higher depolarization ratios than the aerosols from the land. Data from Fukushima observations is expected to follow this trend and can be used to classify which aerosols carry radioactivity. The aerosol concentrations will be measured using aerosol samplers. The data from the samplers, weather

instruments, and lidar can be compared to determine how much aerosols are from the land and the Pacific Ocean. From the

amount of land-based aerosols, the radioactivity level from inhalable aerosols can be regularly assessed. The proximity of the deployment site to the Pacific Ocean indicates that when the dominant wind comes from the ocean, nonradioactive marine-type aerosols dominate. Water vapor concentration is expected to increase, and the observed extinction coefficient will also increase. However, marine-type aerosols should have a lower extinction coefficient and higher depolarization ratio than land-based aerosols. By using $PM_{2.5}$ data, the V-like structure relationship with wind direction can also aid in inferring whether the observed aerosols are land or sea-based.

## 5 Conclusions

This work has shown that the data from continuously operated horizontal lidar, weather monitor, and $PM_{2.5}$ measurements can be used to describe and quantify the relationships among the optical properties of aerosols, weather parameters, and $PM_{2.5}$ concentrations. The results have shown that extinction coefficients of dust and aerosols increase with RH, rain rate, $PM_{2.5}$ amount, wind speed, and absolute humidity. The same parameter decreases with depolarization ratio and wind direction. The lidar-derived depolarization ratio, on the other hand, increases with wind speed and wind direction and decreases with extinction coefficient, RH, rain rate, $PM_{2.5}$ concentration, and absolute humidity. Quantified relationships between lidar-derived parameters (extinction coefficient and depolarization ratio) and weather parameters have been established from the seven-month data providing an understanding of how the optical properties can change with each weather parameter. $PM_{2.5}$ concentrations are observed to increase with RH, but a decreasing mean $PM_{2.5}$ concentration is observed for RH>80%. In previous work, this decrease is observed when RH>70%, implying a different type of aerosols is observed in Chiba.

From these quantified relationships, trends can be inferred, and these can be used to analyze the data when the horizontal lidar is deployed in Fukushima. A similar methodology treatment on the data from continuous observations in Fukushima can be used. The results from the analysis of combined lidar, weather, and sampling measurements can provide an assessment of the air quality and the amount of inhalable radioactive aerosols in the area.

In a radioactive area, wind direction can dictate the level of radioactive aerosols. In Fukushima, when the dominant wind comes from the land, the prevalent aerosols and dust are expected to carry a traceable amount of radioactivity. When the prevailing wind direction is from the Pacific Ocean, marine-type aerosols that do not carry radioactivity will be dominant. Therefore, an increase in $PM_{2.5}$ concentrations with RH does not imply an increase in aerosol number concentration (and an increase in the radioactive level of inhalable aerosols). The increase of extinction coefficient with RH and absolute humidity also does not indicate an increase in radioactive aerosols. The increase of extinction (or the decrease of depolarization ratio) when RH or absolute humidity increases suggests that water vapor is just condensing on radioactive aerosols.

The horizontal lidar is scheduled for deployment to an uninhabited area in Fukushima. The lidar will be operated continuously to detect radioactive aerosol and dust in conjunction with weather instruments and air samplers. The preliminary results from this work have provided qualitative and quantitative insights into the lidar-weather-$PM_{2.5}$ concentration relationship that is valuable in assessing the occurrence of radioactive aerosols and dust in the atmosphere.

Will the data from this measurement be relevant from a global perspective and remote sensing point of view? Ideally, the same instruments must be deployed at different strategic locations to investigate local aerosol-weather relationships at various international locations. The data from this observation and the corresponding results presented in this work characterize and define the local aerosol-weather relationship. The dataset can potentially be used for global climate models to assess results from downscaling experiments.

*Data availability.* Data used in this study are available upon request (shiina@faculty.chiba-u.jp and nofel@civil.kyushu-u.ac.jp).

*Competing interests.* The authors declare that they have no conflict of interest.

*Author contributions.* The project was conceptualized and supervised by TS, YS and TO. Formal analysis, writing and editing of the data was carried by NL. Data curation and system maintenance were done by NL and KO. $PM_{2.5}$ data were curated by HI, YM and TN.

*Financial support.* This research is partly funded by the Japan Environmental Storage and Safety Corporation (JESCO).

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
