# Peer review of "Continuous observations from horizontally pointing lidar, weather parameters, and $PM_{2.5}$ : a pre-deployment assessment for monitoring radioactive dust in Fukushima, Japan"

_EGUsphere, 2023_

## Referee Comment (RC4)

[referee-annotated manuscript omitted]

---

## Author Comment (AC4)

[revised manuscript text omitted]

---

## Author Response (AR1)

Author response to AMT review
(Response to reviewers' comments)
Atmos. Meas. Tech. Discuss., https://doi.org/10.5194/egusphere-2023-546

Continuous observations from horizontally pointing lidar, weather parameters, and PM$_{2.5}$: a pre-deployment assessment for monitoring radioactive dust in Fukushima, Japan

Nofel Lagrosas, Kosuke Okubo, Hitoshi Irie, Yutaka Matsumi, Tomoki Nakayama, Yutaka Sugita, Takashi Okada, and Tatsuo Shiina

Reviewer #1:

State of the aerosol lidar technique is the discrimination between **volume-related** quantities (particles plus molecules) and **particle-related** quantities.

Response:

Lidar systems are constructed to respond to specific scientific goals or objectives. Thus, different lidar systems have pros and cons depending on the scientific questions they are designed to answer. The current lidar system, which is a Mie-scattering lidar, can't discriminate **specific** molecules from particles but is constructed to monitor the smaller particles (not specifically distinguish the types of aerosols) because smaller particles carry larger amounts of radioactivity. Mie lidars are particularly effective at providing insights into the temporal changes of the volume-related quantities, especially when operated continuously, providing information on the changes of volume-related quantities in the order of seconds or minutes. This property is **one of the "state of aerosol lidar techniques"** the authors want to emphasize in this work.

State of the aerosol lidar technique is the discrimination between **different particle types** as dust, maritime, urban, smoke, … Are there **different air mass types**?

Response:

For the data from continuous monitoring, different air mass types can be discerned or inferred from observations. Ancillary weather data show that different seasons have different dominant wind speeds and directions, indicating that different aerosols are being brought in and detected by the lidar system. This observation is a significant contribution that can be used to aid in the analysis of Mie lidar data. This idea is explained in Sec. 4.1 in the manuscript.

As atmospheric measurements are taken, one should (only) think about **relations between quantities for one air mass type**.

Response:

The authors agree with the reviewer on this line of thought. However, in reality, when continuous measurements are performed, the recorded data are the results of the **constant mixing** of air masses and weather's effect on the aerosols' optical properties. Even though Mie lidar data alone can't determine the air mass type, the data from the observation gives information on how aerosol optical properties are affected by changing air masses and weather.

State of the aerosol lidar technique is to **discriminate between intensive and extensive** quantities.

Intensive particle properties are for instance the particle depolarization ratio or the color ratio or the lidar ratio while extensive particle properties are for instance the particle backscatter coefficient or the extinction coefficient.

**=> all is not done in the paper.**

 Response:

Similar to what is stated above, not all lidar systems focus on discriminating the intensive and extensive properties. A lidar system is built to serve a specific scientific goal so that specific scientific questions can be answered. In this system, the discrimination of aerosol types is not an issue that we want to resolve. We want to focus on the small radioactive particles, which are locally transferred or dispersed due to weather and terrain orientation. We evaluate this with lidar and weather instruments. We are also interested in answering the question "What is the relationship between optical and weather parameters?" so that we can quantify the effect of weather on the aerosol optical properties needed to asses radioactive aerosols when the system is placed and operated in Fukushima, Japan for continuous monitoring.

The reader cannot follow the identification of "**radioactive dust**":

Generally: one cannot claim to identify radioactive dust (particles) without measurements of radioactivity at all. The optical (lidar) observations of dust from Fukushima **may or may not** represent radioactive particles.

The reader must be convinced by additional measurements or citations that Fukushima represents a source of only radioactive particles. But the lidar may observe also particles from areas upwind of Fukushima (as maritime …) which are mixed down to the planetary boundary layer between Fukushima and Chiba.

Response:

The authors have added the statements in the introduction section (L 21-33): "Radioactive aerosols are aerosols that have been exposed to nuclear radiation and thus have become radioactive. In a place where the unprecedented release of nuclear radiation brings about radioactive aerosols, these can pose a risk to nearby inhabitants. Past studies have evaluated radiation using gamma-ray detectors and measured radiation dose rates at different sites to analyze radiation values from dust and infer radioactive dust's wind transport (Allott et al., 1992; Takagi et al., 2019; Yamauchi, 2012). These techniques are adequate, especially when evaluating the safety of an area exposed to radiation for people to return Matsuo et al. (2019). However, since the atmosphere is dynamic, wind transports these radioactive aerosols to different places. Monitoring these transport in a higher temporal scale can be an essential observation to assess radioactivity in an area. Remote sensing instruments, like lidars, that can be left alone and gather data with minimal human interference are valuable in monitoring radioactive aerosols. Lidar networks are used to monitor vertical distributions of dust and aerosols in the atmosphere (Campbell et al., 2002; Kawai et al., 2018; Pappalardo et al., 2014) and validate lidar data (Kim et al., 2008; Whiteman et al., 1990; Gusmão et al., 2020). But lidar systems, together with other in situ instruments, can be deployed in radioactive areas and operated continuously with less human intervention and operation. This possibility is the focus explored in this paper.."

I cannot follow the **published extinction values** … my feeling tells me that they are too large … and would yield to low visibilities (Koschmieder Formula). (Okay, I miss the molecular extinction values. The published extinction values represent at least a lot of molecular extinction.)

Response:

The molecular extinction coefficient is high at UV wavelengths. In this paper, the extinction coefficient presented in the paper is the total extinction coefficient.

Fig.1: Are also clouds shown there in? My proposal is to include the direction of the lidar beam (W-NW!) which is not directed to Fukushima. I would include the orography and possibly also the main land cover types (cities, fields, forests) to convince the reader regarding the observed particle types (depol ratio).

Response:

The authors have changed Fig. 1 using NASA's Worldview map. The new figure shows Chiba's location and the possible observation site in Fukushima.

Regarding the presented lidar (Table 1): Does the lidar system have a data acquisition? What are the specs of this sub system?

Response:

The lidar system was operated continuously and gathered data every 1 s, as described in the methodology section of the manuscript. Thus, a data acquisition system is an integral part of the system. The data acquisition system uses a digital oscilloscope programmed to retrieve and store data after accumulating points in one second. The system can have a spatial resolution of 0.375 m and can detect up to a maximum distance of 700m.

Depolarization values (only depol ratio values >= 0.1 [Line 178/179]) are different than values in the text books and request a standard QA/QC.

Response:

The high depolarization ratio results from the lidar being located near the ground and pointed horizontally. As stated in the manuscript, these can be attributed to the observed dust from the ground and snow droplets. To the author's knowledge, continuous horizontal lidar observations have not been performed over a long period. Previously published values of depolarization ratio are obtained from vertically pointing (upward/downward) lidar. Recent and previous published results have shown that measured and modeled depolarization of dust can have values similar to what was observed in Chiba:

https://opg.optica.org/oe/fulltext.cfm?uri=oe-31-6-10541&id=527943
https://acp.copernicus.org/articles/22/355/2022/acp-22-355-2022.pdf
https://agupubs.onlinelibrary.wiley.com/doi/10.1029/2021JD035629
https://www.sciencedirect.com/science/article/pii/S0022407320303502
https://www.mdpi.com/2073-4433/13/12/1946
https://www.tandfonline.com/doi/epdf/10.1111/j.1600-0889.2011.00556.x?needAccess=true&role=button
https://amt.copernicus.org/articles/7/3717/2014/amt-7-3717-2014.pdf
https://www.nature.com/articles/s41598-017-00444-w
https://acp.copernicus.org/articles/17/10767/2017/acp-17-10767-2017.pdf
https://onlinelibrary.wiley.com/doi/full/10.1111/j.1600-0889.2008.00396.x

I could not find a working hypothesis or the physical explanation in the paper why the authors correlated intensive and extensive properties.

Response:

The authors thank the reviewer for this comment. We have added the following sentences in the second paragraph of the introduction section of the manuscript (L. 57-62): "Furthermore, having a dataset of optical properties of local aerosols and quantified aerosol-weather relationships helps understand the local transport of these aerosols, aids in

analyzing radiative forcing, air quality, health impacts on the residents, precipitation patterns, atmospheric dynamics, validating other remote sensing data, and can be used for validating and improving local climate models. Combining these with ancillary data from weather monitors, aerosol samplers, and chemical analysis, monitoring the flow of radioactive aerosols can be analyzed to infer possible locations of hot spots."

I stopped reading at page 11.

Response:

The authors thank the reviewer for the time devoted to reviewing and sharing thoughts on this work. The authors understand that not all contents in this work resonate well with every reader, and we respect the reviewer's decision not to read all the contents and results presented in the manuscript. The authors will continue to improve the manuscript based on the comments from other reviewers.

Reviewer #2:

This study operated a horizontally pointed lidar system over a period of 7 months in conjunction with a weather station and PM2.5 sensor. The lidar was used to observe temporal changes in the optical properties of aerosols; these optical properties were also compared to other measured quantities including RH, absolute humidity, rain rate, wind speed and direction, and PM2.5 concentration. The authors plan to use this knowledge in a followup study in Fukushima, Japan to monitor radioactive dust.

Main Comments:

I felt like the introduction was severely lacking in background information and how this work fits in with previous literature. While a whole paragraph was used to describe radioactive particles the actual study doesn't measure radioactive aerosols. More information needs to be included on lidar, how lidar has been used in previous literature, and other validation techniques that have been used. While Fukushima is brought up, the circumstances surrounding Fukushima are never stated.

Response:

Lidar studies have been used to observe optical properties in the atmosphere, map terrain, and monitor atmospheric events. To respond to this comment, the authors have added several previous works on common lidar networks (MPL, EARLINET and NIES) to provide background on how lidars are used to continuously monitor the atmosphere (L. 32-33).

In this paper, the authors are introducing the idea of using continuously operated lidar systems deployed in radioactive areas with minimal human operations. To introduce this

idea, the authors added the statement (L. 34-35): "But lidar systems, together with other in situ instruments, can be deployed in radioactive areas and operated continuously with less human intervention and operation. This possibility is the focus explored in this paper".

The circumstances surrounding Fukushima have now been stated. The authors have added the statements (L. 41-44): "After the radiation incident in Fukushima area in 2011, places near the radioactive meltdown have become uninhabited (Hidaka et al., 2022). However, years after the incident, radioactivity has declined. Still, many of the inhabitants have a hard time returning due to various reasons such as absences of family members and neighbors, and insufficient radiation decontamination, among others (Matsuo et al., 2019)."

There are many references to radioactive dust but it seems that the radioative dust is never defined until line 264 where it is stated that "sources of radioactive aerosols are the aerosols from the land". This should be stated more up front. Additionally, more information should be included regarding the actual definition of radioactive dust; is this based on some radioactivity level limit or the aerosol composition?

Response:

The authors have defined radioactive in the first paragraph of the introduction (L. 21-22). The authors have defined these aerosols/dust in a broader way to encompass anything that has been affected by nuclear radiation.

Line 270: It is stated that "from the amount of land-based aerosols, the radioactivity level from inhalable aerosols can be assessed." Where is the evidence for this? There seem to be at least 2 assumptions in this statement; 1: that all land-based aerosols in Fukushima are radioactive and 2: that the radioactivity levels of aerosols can be derived from lidar data? Wouldn't #1 lead to an overestimation of inhalable radiotion exposure?

Response:

Radioactivity in Fukushima has been decreasing through the years. The authors have added this information from the works of Hidaka et al., 2022 and Matsuo et al., 2019 (L 39-42) as "evidence". Thus, not all of the land-based aerosols in Fukushima area are radioactive. This means no overestimation of inhalable exposure.

By having the soil analysis, the amount of radiation from dust can be measured. Lidar data alone can't derive radioactivity but can aid in analyzing radioactive aerosols and dust (optical, amount, movement, etc.) from combined meteorological, remotely sensed and soil analysis data.

Section 3.2 Please provide more information on the position of the weather station/CEReS building relative to the lidar/Engineering building.

Response:

The authors have revised the second sentence of the first paragraph under Sec. 3.2. The revised sentence now reads (L. 110-111) "The weather monitor is around 74 m horizontally and 19 m from the lidar system. It logs data on weather parameters (RH, temperature, wind speed, wind direction, and rain rate) every 5 min.

Section 4.1 It is mentioned that optical parameters depend on seasons and that different aerosols will be present in the summer compared to winter yet your study does not cover several spring and summer months, therefore missing crucial data to further quantify the relationship between lidar optical properties to weather and PM2.5.

Response:

It is true that the lidar data cover only seven months (one summer month, three autumn and three winter months). However, our previous long-term measurements (Fukagawa et al., 2006) using air samplers, chemical analysis, weather data and sun photometer have shown that aerosol properties are dependent on seasons. To clarify this, the authors have added the statement (L. 128-129) "This seasonal dependence was also observed in previous studies using a chemical sampler, chemical analysis, weather data, and sun photometer." in the first paragraph of Sec. 4.1.

Figure 3 shows the mean RH for 3 months was 80%, which is above the PM2.5 sensors threshold as demonstrated in Nakayama et al. 2018. As stated in their study at RH above 70% the sensors overestimate PM2.5. This ties in to line 206 where an increase in PM2.5 concentration is observed during increases in RH. Could this just be an artifact of the sensor overestimating PM2.5? This would also impact the comparison of sensor PM2.5 and lidar optical properties at RH>70%.

Response:

In Nakayama et al., 2018, the PM2.5 instrument was compared with standard instruments (DKK-TOA, FPM-377 and Kimoto, PM-712). The 70% "threshold" mentioned in the manuscript represents the situation when hygroscopic growth does not influence the measurements. It should be noted that the standard instruments used to compare PM2.5 concentrations are heated systems (DKK-TOA, FPM-377) or have filter samples maintained at specific temperature and RH (Kimoto, PM-712). Thus, the PM2.5 concentration values observed for RH>70% have the inherent effects of aerosol growth and can have overestimated values with respect to heated systems. When optical parameters from lidar are compared with PM2.5, the hygroscopic effect, observed at higher PM2.5 values, deviates from the trend as shown in Figs. 5b and 5c. To resolve this, the authors have added the statement (L. 245-246) "At RH>80%, the mean PM2.5 concentration

deviates from the trend. This observation can be attributed to the fact that the PM2.5 values for high RH (>70%) are affected by hygroscopic growth (Nakayama et al., 2018)."

Line 268 In this study a small PM2.5 sensor was used to quantify PM2.5, yet in your future study you propose a different method, a "dust sampler". How can you expect the conclusion you drew between the lidar and PM2.5 sensor to be the same for lidar and a "dust sampler"?

Response:

The authors have used the term "dust sampler" to mean a device that measures fine and coarse aerosols. To convey this meaning, the authors have replaced the word dust with aerosol in L. 311.

The results from the continuous observation in Chiba have provided us with knowledge of the trend between the PM2.5 sensor and lidar data. We don't expect the same trend to exist since the measurements in Fukushima will be near the ground (~1 m). However, we expect a similar trend to exist.

Minor Comments:

Line 101 There seems to be an extra space after m$^{-1}$

Response:

Thank you for noticing this. This has been corrected.

Line 206 Not sure why this implication specifically applies to areas with radioactive aerosols.

Response:

The results of the observations done in Chiba produce quantifiable relationships between optical and physical quantities. These relationships may differ slightly at different places, but such trends should exist whether the aerosols are radioactive.

Line 222 "due to the effect of RH becoming PM2.5 particles" This does not make sense, are you referring to the growth of aerosols due to deliquescence?

Response:

The authors mean the growth of aerosols due to deliquescence. The authors have revised this statement to "Increased PM2.5 concentrations indicate aerosol growth by

deliquescence, i.e., formation of more near-spherically shaped aerosols with sizes less than 2.5 μm." (L 260-261)

Reviewer # 3:

The title of the study "Continuous observations from horizontally pointing lidar, weather parameters, and PM$_{2.5}$: a pre-deployment assessment for monitoring radioactive dust in Fukushima, Japan" by Lagrosas et al., implies that a horizontally pointing lidar together with basic atmospheric weather measurements can be used to monitor radioactive aerosol particles in Fukushima. The study presents 7 months of measurements in Chiba. By means of monthly statistics and correlations between lidar (extinction and depolarization) measurements with PM$_{2.5}$ and weather parameters (e.g. relative humidity, wind) a separation between land and marine aerosol particles is made. The authors state that the aerosol particles originating from the land carry the radioactivity and, hence, can be used to derive aerosol radioactivity levels in Fukushima.

In my opinion, this measurement setup and simple analysis approach do not meet the scope and quality standards of AMT. While I can see the general idea of having a continuous monitoring of the aerosol particles to warn residents of potentially higher radioactivity levels, the paper substantially lacks an introduction and discussion of the past 60 years of boundary layer aerosol and radioactive particle research. Hence, I cannot recommend this study for publication in its current state. Below you find the general and specific comments that led to my overall recommendation.

Response:

The main idea presented in this study is to use continuously operated remote sensors with less human interference, such as lidar (and other instruments such as weather monitors and aerosol samplers), to be placed in radioactive areas to monitor radioactive aerosols. Previous research on radioactive aerosols uses only gamma-ray detectors to detect radioactivity in an area. To respond to the reviewer's comment, the authors have added the following sentences in the introduction (L. 21-35): "Radioactive aerosols are particulates that have been exposed to nuclear radiation and thus have become radioactive. In a place where the unprecedented release of nuclear radiation brings about radioactive aerosols, these can pose a risk to nearby inhabitants. Past studies have evaluated radiation using gamma-ray detectors and measured radiation dose rates at different sites to analyze radiation values from dust and infer radioactive dust's wind transport (Allott et al., 1992; Takagi et al., 2019; Yamauchi, 2012). These techniques are adequate, especially when evaluating the safety of an area exposed to radiation for people to return Matsuo et al. (2019). However, since the atmosphere is dynamic, wind transports these radioactive aerosols to different places. Monitoring these transport in a higher temporal scale can be an essential observation to assess radioactivity in an area. Remote sensing instruments, like

lidars, that can be left alone and gather data with minimal human interference are valuable in monitoring radioactive aerosols. Lidar networks are used to monitor vertical distributions of dust and aerosols in the atmosphere (Campbell et al., 2002; Kawai et al., 2018; Pappalardo et al., 2014) and validate lidar data (Kim et al., 2008; Whiteman et al., 1990; Gusmão et al., 2020). But lidar systems, together with other in situ instruments, can be deployed in radioactive areas and operated continuously with less human intervention and operation. This possibility is the focus explored in this paper."

General comments:

The paper is well-structured and written, but the quality of the figures is poor: Fig. 1 is blurry and might be a satellite image, but no source is provided. In Figs. 2-7 the axis titles are very small and blurred when zooming in. In Fig. 4b) and h) the y-axis title is missing, and I cannot read the y and $r^2$ values.

Response:

The authors have replaced a new map obtained from NASA's WorldView map.

The authors have improved the quality of the figures. The overlapping figures have caused the y-axis to be concealed. The authors have now fixed this problem. The fonts (including the equations) have been enlarged.

Also the authors have corrected the equation in Fig. 4b to $y = 0.001735 * \exp(0.00477x)$ from $y = 0.001654 * \exp(0.5028x)$, where the variable $x$ in the former is the RH while the variable $x$ in the latter is RH/100. The slight correction presents no changes in the discussion.

In the introduction, an explanation and definition of aerosol particle types is missing. Please also provide some background about their origin and potential seasonality that can be expected in the boundary layer in general and in particular for the sites in Chiba and Fukushima. Throughout the manuscript terms, such as aerosols, dust, urban-type, marine-type, hydrophilic aerosol, regular aerosols are used, but which types may become radioactive? Please also provide information on typical particle sizes of radioactive particles. What about larger particles? Why didn't you include $PM_{10}$? Did you ever measure (externally confirmed) radioactive aerosol particles with this lidar?

Response:

In the introduction, the authors have added the statement (L. 16-17): "Aerosols are solid and liquid particles suspended in air and are composed of organic compounds, inorganic salts, trace elements, black carbon, water and other substances (Reggente et al., 2019)."

The authors also added the following sentences on potential seasonality

1. (L. 19-20): "Other studies have shown that aerosols and their optical properties in a particular area show seasonality (Humphries et al., 2023; Orikasa et al., 2020; Jain et al., 2020; Cigánková et al., 2021)."
2. (L. 39-40): "In Chiba, the seasonality of aerosols has been reported (Fukagawa et al., 2006), while in Fukushima, the seasonality of aerosols is still to be quantified."

When nuclear radiation happens, everything in its way can be contaminated. Thus, all possible types of aerosols can be potential radioactive aerosols. For this reason, the authors have added the definition of radioactive aerosols in L. 21-22: "Radioactive aerosols are particulates that have been exposed to nuclear radiation and thus have become radioactive."

Our previous studies have shown that dust smaller than 20 μm in diameter carries more radioactivity ($\sim 2 \times 10^{-8}$ Bqcm$^{-3}$) than larger dust (Shiina et al., 2018). This is already stated in L. 46-48.

In Chiba, the authors are limited to using just PM2.5 data since PM10 instruments are not available near the observation site. However, in future observations in Fukushima, samplers including PM10 will be used.

Our group did measurements of radioactive soils in Fukushima combined with aerosol samplers. This was reported in Shiina et al., 2018. Lidar alone can't measure radioactivity. Radioactivity is measured using aerosol samplers and chemical analysis of aerosols. If the samplers, weather monitor, lidar, and other instruments are operated continuously, the combined data will reveal sources of radioactive aerosols, optical changes of these radioactive aerosols, among others.

I'm not convinced that the aerosol particle characteristics of the station in Chiba on the 9th floor of a building (65 m above sea level), which is in a metropolitan area with industry and traffic, is comparable to a rather rural station 1 m above the ground in the mountains of Fukushima province. While the station in Chiba is close to the sea, about 3.5 km away from the Bay of Tokyo and 33 km away from the Pacific Ocean, where will it be in Fukushima? The city of Fukushima is about 45 km away from the sea and 65 km away from the nuclear power plant. Why do you expect that air from the sea will not take up some radioactive material on its way from the coast to the city of Fukushima? Is there any agriculture where tilling can lift mineral dust? What about biogenic aerosol?

Response:

The aerosols particle characteristics in Chiba are definitely different from that of the observations in Fukushima in the future. What this manuscript is proposing is the possible determination and quantification of optical characteristics of aerosols in Fukushima based on the methods and techniques applied to the data observed in Chiba.

Currently, the plan is to place the lidar and other instruments in between uninhabited and inhabited area in Fukushima. The specific place will be decided in the future.

During the nuclear event in Fukushima, the contaminated areas are the land. If there is radiation from the sea, this could be due to some transport from the land, since the radioactive water from the power plant was already stored.

Agriculture in Fukushima is done in safe areas where radiation is very minimal. What is crucial here is the dust from uninhabited areas that can be transported to inhabited areas. For this reason, we plan to put the system between these two areas.

Currently, radioactive biogenic aerosols are not the focus of this work. If such aerosols exist during the observation in Fukushima, they will be known from the sampling, and their radiation level can be determined after chemical analysis. This will be one of the questions that we can answer in the future.

A discussion of the results is missing. Please compare your results with findings in literature and discuss the implications of your conclusions, e.g. the relation between $PM_{2.5}$ and RH (Lou et al. 2017).

Response:

The authors have added the statements (L. 245-248): "At RH>80%, the mean PM2.5 concentration deviates from the trend. This observation can be attributed to the fact that the PM2.5 values for high RH (>70%) are affected by hygroscopic growth (Nakayama et al., 2018). Previous works have observed this deviation at RH=70% (Lou et al., 2017; Wang and Ogawa, 2015; Zalakeviciute et al., 2018) and have attributed this to the effects of rain, hygroscopic growth that leads to reducing PM2.5 concentrations".

The authors have also added the statement (L. 327-329) in the conclusion:" PM2.5 concentrations are observed to increase with RH but a decreasing mean PM2.5 concentration is observed for RH>80%. In previous work, this decreases is observed at RH=70% implying a different type of aerosols observed in Chiba."

Specific comments:

l24: Does the number indicate the absolute radioactivity, or how much more? Please specify. What is the natural background radioactivity? Please provide a value for comparison. Please specify if 20 μm is the particle radius or particle diameter.

Response:

The unit Bqcm$^{-3}$ is the measurement of radioactivity per cubic centimeter (radioactivity density) of space occupied by a particle, which is in this case, comes from the soil. The absolute radioactivity is the product of the radioactivity density and the volume. Since the size of dust varies, the authors prefer to report the radioactivity density.

The 20 μm mentioned in the manuscript refers to the size of the dust and therefore refers to the diameter. To clarify this, the authors have added the phrase "in diameter" in L. 46.

l46: Please provide more details for the planned measurement station in Fukushima. Fig. 1 indicates that it is not directly at the coast, but the text suggests that it is next to the Pacific Ocean.

Response:

As mentioned the specific details have not yet been decided. The observation site will be in between uninhabited area and inhabited area. To address this, the authors have added the statement (L. 74-76): "The specific site location will be decided in the future but the site will be in between uninhabited and inhabited areas."

l51: Please specify which device is meant with "weather monitor".

Response:

The weather monitor is a Davis Pro2 weather monitoring station. This is information is now added in L. 109.

p4: What is the detection sensitivity of the lidar? What is the lowest extinction you can discern from molecular scattering, what is the highest extinction coefficient? Can radioactive aerosol loads below your detection limit become harmful to humans and animals?

Response:

The lidar's sensitivity depends on the detector. The sensor, which is a PMT operated in photon counting mode, has a cathode radiant sensitivity of around 100 mA per 1 W of light at 349 nm. When operated at high frequency to gather data per second, the minimum observed extinction coefficient is 7.9 x10$^{-4}$m$^{-1}$ which is comparable to the molecular signal at 349 nm. The highest observed extinction coefficient is 0.0144m$^{-1}$. The radioactive loads can be observed using chemical analysis of sampled aerosols, which can later be compared with lidar data. The minimum value of the extinction coefficient comes from the gas molecules, i.e., no aerosol loading. We don't expect any radioactivity that can be harmful to humans and animals to come from gas molecules.

p5, Table2: The text states that the lidar is only stopped for a few minutes for a reset. Why are so many days missing? Did you apply some filtering, for e.g. clouds or fog?

Response:

The system was started in the middle of August 2021. During heavy rains, typhoons (common in Japan in October) and maintenance, the system must be stopped to prevent damage to the laser and the optics. No filtering was applied for clouds or fog.

l79: How far away was the weather station? Please specify the uncertainties of the instrument and due to the distance to the lidar.

Response:

The authors have added the statements (L. 110-111): "The weather monitor is around 74 m horizontally and 19 m from the lidar system. It logs data on weather parameters (RH, temperature, wind speed, wind direction, and rain rate) every 5 min".

Since the weather monitor is close to the lidar system, uncertainties due to the distance are expected to be very negligible.

The weather monitor, which is a Davies Vantage Pro2, is a widely used weather monitor around the globe. This instrument is known to produce accurate results (Jenkins, 2014). In our group, the weather monitor has not been observed to malfunction during the course of the seven-month operation. For this reason, any uncertainties from the weather monitor is none or, at most, negligible.

- Jenkins, G. (2014). *A comparison between two types of widely used weather stations*, Weather (Royal Meteorology of Science), 69, 4, 105-110.

l85: Why do you only include $PM_{2.5}$ and not $PM_{10}$? Your introduction indicates, that particles smaller than 20 μm are important. Please provide a reference for SKYNET.

Response:

PM10 instruments are not available at the Chiba site. However, during the observation in Fukushima, PM10 data will be available.

The authors have added 2 references (Nakajima et al., 2020 and Hashimoto et al., 2012) for SKYNET (L. 117-118)

l94: Please comment on the advantages and disadvantages of the robust fitting method used in this study.

Response:

Using robust fitting in analyzing data has the advantage of resilience to outliers, increased reliability, and general applicability. On the other hand, using robust fitting can result in computational complexity and loss of efficiency. However, these problems were not encountered in analyzing the data. These ideas are added in the methodology section L. 85-86.

l96-97: I find it a bit speculative stating that the optical parameters depend on season, if not even a full year was measured. Can you provide references that support this claim? Is this decrease significant?

Response:

Optical parameters depend on the season. These results have been published in the following papers:
https://www.sciencedirect.com/science/article/pii/S1352231009004117
https://acp.copernicus.org/articles/11/10661/2011/
https://agupubs.onlinelibrary.wiley.com/doi/full/10.1002/2014JD021500
https://www.sciencedirect.com/science/article/pii/S1352231005011660
https://journals.ametsoc.org/view/journals/atot/28/10/2011jtecha1532_1.xml

Based on these previous results, we expect that optical parameters observed by lidar must depend on seasons. Based on physics, a simple explanation is that during a month with high relative humidity, we expect water vapor to condense on aerosols, making them bigger. The increase in size produces changes in the extinction coefficient. Thus, if one keeps on getting data during this month, the average of the optical parameter will be different from that of a dry month. From this, the authors are confident that the decrease in extinction coefficient presented in Figs. 2a is significant.

l105-106: Were the extinctions found in previous works measured at the same location? Were any pollution reduction plans implemented since these measurements? Were these measurements made during comparable meteorological conditions?

Response:

The extinction coefficients measured in previous works (Ong et al., 2019 and Xiafukaiti et al., 2020) are observed in the same location but using different lidar systems. The measurements are **continuous**. Thus, these measurements are made at all possible meteorological conditions except during heavy rains.

Pollution reduction has always been active, monitored and implemented in Japan using different sensors placed at strategic locations (https://soramame.env.go.jp/station, in Japanese).

l115: Please provide a reference for the depolarization ratios attributed "dust".

Response:

The authors have added two references (Haarig et al., 2022 and Huang et al., 2023) (L. 151-152).

l145-147: How do you know that this is dust? What do you mean with dust? Mineral dust? Combustion aerosol? Ice crystals?

Response:

The dust mentioned in this work means collective dust coming from the ground since the observation is near the ground.

l187-188: What is the added value of relating the extinction coefficient with the absolute humidity?

Response:

A relationship between the extinction coefficient and absolute humidity provides quantitative information on the amount and how water vapor affects the extinction coefficient, especially during humid conditions. This information is essential for modeling optical parameters and the scattering effects that take place from these optical changes.

l220: What does "regular aerosol" mean here?

Response:

The authors mean the observed local aerosols. To clarify this, the authors have changed the word "regular" to "observed local" (L. 263)

l221-222: Do you mean particle formation occurs here?

Response:

The authors mean aerosol growth by deliquescence. The authors have edited the sentence to (L. 264-265): "Increased PM2.5 concentrations indicate aerosol growth by deliquescence, i.e., formation of more near-spherically shaped aerosols with sizes less than 2.5 μm."

l226: I find it difficult to see a V-like structure. There are so many overlapping data points. A box-whisker-plot may be more useful in Fig. 5d,e).

Response:

The authors have changed Figs 5d and 5e to an error bar plots.

l235-240: Same for Fig. 6. I can't see any pattern that could lead to interpretation drawn in the manuscript. Are the discussed signals significant? Minimum and maximum values are out of plot boundaries. How do you know that the "two peaks" can be attributed to different aerosol particle types? The mean of the depolarization does not show a similar pattern. What type are the "naturally occurring aerosols" and what is the difference to marine-type and dust?

Response:

The discussion pertaining to Figs. 6a and 6b refer to the trends of the average value of the extinction coefficient and depolarization ratio. This is stated clearly in the first sentence of the paragraph (L. 278). To clarify this point, the authors have added the word "average" before the words "extinction coefficient" and "depolarization ratio" in this paragraph and the succeeding paragraph (L. 278-289). Furthermore, high error bars are expected when one deals with continuous observations since the atmosphere is dynamic in nature.

The attribution of the aerosol type to each peak flows naturally from the fact that if the wind speed is near zero, the aerosol type that can be observed can be only near the lidar system and not coming from far locations. For high wind speed conditions, the aerosols that can be measured can have contributions from sources far from the lidar system.

References:

Lou, C., Liu, H., Li, Y. et al. Relationships of relative humidity with PM2.5 and PM10 in the Yangtze River Delta, China. Environ Monit Assess 189, 582 (2017). https://doi.org/10.1007/s10661-017-6281-z
**Citation**: https://doi.org/10.5194/egusphere-2023-546-RC3

Reviewer # 4:

The authors present a study that links aerosol optical properties and PM2.5 mass concentrations measured from a lidar system and a particle sample with meteorological

parameters. The measurements are performed in a 7 month period and the analysis is done with the prospect of being applied in the radioactive environment of Fukushima.

While the analysis of the seasonal aerosol patterns and their link to weather conditions is quite solid, the connection to the detection of radioactive particles is not well established. I would recommend the publication of this paper if the radioactivity part is removed in the title and most of the manuscript but mentioned in the future plans. The study already includes rather interesting aerosol-related findings on its own, it does not have to be tied aerosol radioactivity that is anyway not well explained in the discussion. Are only dust particles expected to be radioactive? What about the fine mode? These are all questions for a future publication after the system is moved to Fukushima. For now, I would recommend to stick to what is already there in terms of measurements.

Response:

The reviewer has suggested that the "radioactivity part" of the manuscript be removed. The authors have thought about this, and we have listed down the reasons why the "radioactivity part" should be kept in the manuscript:

1. The main objective of this work is to show that lidar systems (together with ancilliary instruments) **can** be used to monitor, study, and gather data on radioactive aerosols as a pre-deployment study. Although the current manuscript does not have radioactive aerosol measurements, the methods and data presented in the manuscript are essential in relating the results when measuring radioactive aerosols and dust. Thus, the authors feel that removing the radioactivity part will reduce the manuscript value to a routinary aerosol study. The authors believe that by retaining the radioactivity part, the whole approach integrates a greater story and impact on aerosol research.
2. The result of the seven-month data presented in the manuscript is already solid in aerosol detection and remote sensing. However, as a pre-deployment study, the authors must address how the current interpretation of the results relates to radioactive aerosols. The current interpretation of the result in relation to radioactive aerosols, we believe, is an added value of information that needs to be put into consideration when performing observations in Fukushima and to adhere to the journal's theme and objectives.
3. The authors believe that the "radioactivity part" in the manuscript serves as a "guide" or a way to interpret the collected data in Fukushima. By stating this in the manuscript, performing data analysis obtained in Fukushima will not start from zero but will be focused on more analysis, e.g., combining the results with transport modeling, local climate model, etc.

More specific comments and corrections are included inline in the pdf supplement.

Aerosols can be transported..

Response: The first paragraph has been edited to include the definition of aerosols and the effect of weather on the optical properties of these aerosols. The first paragraph now reads (L. 16-20): "Aerosols are solid and liquid particles suspended in air and are composed of organic compounds, inorganic salts, trace elements, black carbon, water, and other substances (Reggente et al., 2019). Aerosols can be transported from their sources, which include natural (e.g., volcanos, dust, ocean) and nthropogenic emissions to different locations and are susceptible to environmental and weather conditions. Other studies have shown that aerosols and their optical properties in a particular area show seasonality (Humphries et al., 2023; Orikasa et al., 2020; Jain et al., 2020; Cigánková et al., 2021)."

Transportation is not necessarily affecting the optical properties of individual aerosol species. I would leave this sentence out because it not so well connected to what comes before and after it.
Response:  This statement has been deleted and edited. Please see responses in the first comment.

Please rephrase because this lidar is not introduced yet. For example: In a previous study (Shiina et al., 2018) we have oparated a horizontal lidar near...
Response:  In the current version, lidar has been introduced before (L. 30) this statement.

A citation is needed here
Response:  The work of Shiina et al., 2018 has been added (L. 47).

Is this value of accumulated exposure considered safe? It would be interesting for the readers to provide some more information here from the literature
Response:  This value is considered safe. A new statement (L. 50-54) is added to convey this message and the authors have added reference: "Previous studies indicate that inhaled radioactive aerosols under a dust concentration of $1 \times 10^{-4}$ gm$^{-3}$ can have an effective dose of just 5.6 µSv received in 20 years by an adult with light activity. This dosage is smaller than the total effective dose of 9.9 µSv and 48.8 µSv for adults with moderate and vigorous activities, respectively, during the same number of years at an air dust concentration of $1.5 \times 10^{-4}$ gm$^{-3}$" (Hanfi et al., 2021; Valentin, 2002)

to present continuous data from...
Response: The phrase is now changed to (L. 64): "to present continuous data from a near-ground horizontally pointing lidar system,"

in the

Response: "the" is added before the word "optical" (L. 65).

local aerosols
Dust is part of the aerosols so it is reduntant to be mentioned again separately
Response: The authors have deleted the word "dust" (L. 65).

the observed aerosol optical properties and other meteorological parameters (e.g. ...)
Response: The authors have edited the phrase as the reviewer has suggested (L. 66).

please add also the height above ground level. Is the building located at sea level?
Response: The information of the height above the ground is added (L. 69).

is looking at
Response: The phrase is edited as the reviewer suggested (L. 75-76).

are surrounded by landmasses
Response: The phrase is edited as the reviewer suggested (L. 76).

is covered by both land and sea
Response: The phrase is edited as the reviewer suggested (L. 76).

are expected to be affected from...
Response: The phrase is edited as the reviewer suggested (L. 77-78).

Is it possible to have a higher-resolution and preferably colored image here? While the sea is visible already, it is difficult to descern the mountains and the cities here
Response: The authors have changed the map using NASA's worldview map.

a horizontal lidar, a weather monitoring station, and a particulate matter (PM2.5) sampler
Response: The authors have changed the phrase as the reviewer suggested (L. 80).

meteorological
Response: The authors have change the word weather to meteorological (L. 82).

is pointing at
Response: The authors have edited the phrase as suggested by the reviewer (L. 91).

This sentence is somewhat irrelevant in this section where the lidar is described. I would leave it out

Response: The authors have deleted the sentence as suggested by the reviewer.

Are measurements within the overlap region being used? This information has to be included here. In addition, if the overlap region is used forthe retrievals then the authors must include a paragraph discussing the applied overlap correction method.
Response: The signals before the full overlap region is not used in the analysis of the data. Only the data from 100 m to 300 m are used in the analysis (L. 100).

Is the lidar eye safe? This should be mentioned also here
Response: The authors have added the sentence "The transmitted beam is expanded to 30 mm in diameter and is considered not safe at near distance." (L. 93-94)

Please add also the laser beam divergence and the focal length here
Response: The authors have added the beam divergence information in Table 1.

Usually the maximum pulse duration is provided instead of the minimum
Response: The authors have changed the pulse duration value to 5, the maximum value.

How is the polarization calibration performed? The authors have to provide some additional information here including references
Appart from the gain calibration, it is possible that the receiver optics introduce polarizing effects such as diattenuation and/or retardation depending on the optical setup. In addition, the laser beam is never totally linearly depolarized at the emission and the emitting optics (e.g. beam expanders, protective windows) can sometimes indtroduce additional depolarization.
These effects can be accounted for using the methodology of Freudenthaler et al. 2016. Did the authors apply such a corection? If not, have they verified that the system does not introduce any polarizing effects in the emitting and the receiving part?
Freudenthaler, V.: About the effects of polarising optics on lidar signals and the $\Delta 90$ calibration, Atmos. Meas. Tech., 9, 4181–4255, https://doi.org/10.5194/amt-9-4181-2016, 2016.
Response: In practice, the depolarization ratio of the beam at the exit can be checked using a polarizer and and a power meter. The authors have done this in the course of the calibration method. Thus, whatever polarizing effects contributed by the optics is already accounted in the calibration process. For this reason, the method by Freudenthaler 2016 is not deemed necessary. The approach used by the authors is also similar to the approach used by the NIES lidar team in measuring the depolarization ratio.

is this the volume or particle depolarization ratio? This must be mentioned in the text and the figures as the volume depolarization ratio includes also the depolarization coming from the molecules.

Response: As defined in the manuscript, the depolarization ratio is the ratio of the S and P signals (L. 100). Therefore, the depolarization ratio in the manuscript is the linear depolarization ratio. To respond to the reviewer's comment, the authors have added the word "linear" in L. 104.

This difference is quite large, especially if it is systematic. With 4° inclination at 5km range the signal orriginates from ~350m height. Are there other studies supporting that the aerosol extinction and/or concentration can vary within one order of magnitude in the first 300m height? In addition how does this correlate with columnar sunphotometer measurements and measurements from vertically-pointing lidars (if there is any co-located).

Response: The factor of 10 stated in the manuscript is a rough estimate of the difference. In the previous work, the first 350m height also corresponds to 4.9km in horizontal distance. Thus, in the previous work, the measured extinction coefficient incorporates the effects of this horizontal distance. In the current work, the horizontal distance near the ground is just 300m. Thus, all the laser encounters more particles than if it were directed vertical or 4 degrees above the horizontal. To have a clear picture of this, consider the extinction coefficient data presented in Figure 2a in https://amt.copernicus.org/articles/11/3031/2018/ or the data in Figures 7a, 9a and 10a in https://aaqr.org/articles/aaqr-18-07-oa-0267. The current work is just a point near the ground in these graphs while the previous work can be thought of as extinction coefficient near from the ground to higher altitude, say 6km. The average of these extinction coefficient is clearly smaller than the extinction coefficient near the ground.

As of the moment, the authors have not yet done a comparative study of the current data with existing sun photometer and vertical lidars. However, comparing the two instruments will have dissimilarities since the vertical lidar and the sunphotomter does not see only aerosol information from the ground.

How was this verified?

Response: We have verified this by plotting the histogram (not shown in the manuscript) of the daytime and nighttime depolarization ratio.

is there contribution from other non-spherical aerosol components expected to affect the site (e.g. pollen and/or volcanic ash)?

Response: The site is near the ground and no nearby volcanic eruption occurred during the observation period. Pollens usually occur during spring time. Thus, to the authors' knowledge, the source of non-spherical aerosols during the observation period are from dust, sea salt and snow. We have added this information in L. 160-162: "It is worth mentioning here that during the observation period, no nearby volcanic eruption occurred during the observation period. Pollen usually occurs during spring time. Thus, to the authors' knowledge, the source of non-spherical aerosols during the observation period are from dust, sea salt and snow.".

During winter the mean depolarization ratio increases to 0.2 which corresponds to dust-dominant conditions. Is this local dust? Are other dust sources (e.g. Mongolian dust or desert dust in general) expected to contributed during winter in the region?

Response: The authors attribute this the combination of dust from the ground when the wind is strong and snow. The yellow sand event from China usually occurs in spring time.

Appart from the size, the sphericity is also expected to change and this directly affects the depolarization ratio. Water uptake results to more spherical particles as the water covers or dilutes the condensation nucleus. Please considere adding some relevant references here that support the findings

Response: The authors have added additional references (L. 175)

This part in combination with Fig. 4 a) is not clear. In Fig. 4 a) the depolarization ratio is ploted along the extinction coef., not the RH as the text implies.
In addition, there is a category named "Dust and Snow". it is not clear in the figure if the black lines extend below the orange lines. Please consider using semi-transparent lines. The authors have also to specify what exactly does Dust mean here and how the classification was performed. Was the PM2.5 sampler somehow deployed?
The depolarization ratio of the "Aerosols" category goes as high as 0.4. This is not achievable without dust particles in the mixture. This means that the "Aerosols" category also contains dust particles which creates confusion.

Response: The authors made a mistake here. The RH in the statement should be extinction coefficient. The authors have changed this.
Figure 4a is now edited based on the reviewer's suggestion.
In this work, only a PM2.5 instrument was available. As mentioned in the previous comments, the possible source of dust would be from the ground. We have added this information in L. 160-162.
The reviewer has made a point here. The low extinction coefficient and high depolarization ratio in Fig. 4a indicates the effect of salt. We have added a statement to clarify this (L. 180-181): "The high depolarization ratio and low extinction coefficient can be attributed to the contribution of marine type aerosols as will be shown later when these optical parameters are compared with wind direction. ".

because when

Response: The authors changed since to because (L. 192).

This behavior is expected. Please consider adding some references that show how the hygroscopic growth affects the extincion cross section of aerosols

Response: The authors have added references as suggested by the reviewer (L. 194).

has a higher growth rate?

Response: The authors mean "larger". The authors have edited the sentence to (L. 197-198): "For example, at 90% RH, the extinction coefficient and growth rate of marine-type aerosols are higher than dust Zieger et al., 2013)."

The paper from Haarig et al. 2017 is indeed focusing on pure marine layers. The author can considered using optical libraries (e.g. OPAC), aditionally, to see whether their results are as expected for water soluble aerosols or continental and/or urban mixtures that are probably expected at the site
Response: The authors are interested to consider OPAC in the future. For the current manuscript, additional references are included (L. 201-203). The authors believe that this is sufficient for the time being.

Coated dust with water is not the only and probably also not the most efficient process that can result to a depolarization ratio reduction with the increase of RH. Please consider adjusting the text. Even in dust dominant conditions there is always a non-dust fine mode present that consists of certain amount of non-depolarizing water soluble particles (sulfates, nitrates, organics) (see Hess et al. 1998). During water uptake their size and cross-section increases making their contribution in the mixture much more pronounced considering also their much higher original number density since they are in the fine mode. The result is a decrease of the depolarization ratio as the enlarge water solubles become the dominant component.
Hess, M., Koepke, P., and Schult, I.: Optical properties of aerosols and clouds: The software package OPAC, B. Am. Meteorol. Soc., 79, 831–844, 1998.

Response: The authors have revised the statement to: "Furthermore, the 0.1 average depolarization ratio indicates the limiting value for near-ground aerosols and implies that they cannot possess a perfectly spherical shape even when coated by water at high RH." (L. 217-219).
The authors have also changed the word "dust" in the next sentence (L. 219) to "aerosols".

Please add the missing y-axis label in b) and the non-visible y-axis label in h)
Pleas use always the same units for the same variables (e.g. the RH in x-axis is different in c) and d)
Response:
The authors have improved the graphs and have incorporated the reviewer's comments.

Do all points in figure 4 g) and h) correspond to the "Dust and Snow" points in a)? This is not totally clear here. In that case it would help to add the same legend ("Dust and Snow") also to g) and h) for clarity

Response: The points in Figs. 4g and 4h correspond to all extinction coefficients and depolarization ratio compared with rain rate.

aerosol number concentration

Response: The authors have inserted the word "number" as the reviewer suggested (L. 337).

---

## Author Response (AR2)

Author response to AMT review
(Response to reviewers' comments)
Atmos. Meas. Tech. Discuss., https://doi.org/10.5194/egusphere-2023-546
Continuous observations from horizontally pointing lidar, weather parameters, and PM2.5: a
pre-deployment assessment for monitoring radioactive dust in Fukushima, Japan
Nofel Lagrosas, Kosuke Okubo, Hitoshi Irie, Yutaka Matsumi, Tomoki Nakayama, Yutaka Sugita,
Takashi Okada, and Tatsuo Shiina

Reviewer 1:

The manuscript has been improved a lot after the revisions and the critical missing link between the current analysis and radioactivity seems to be better established. Consequently, there is no need to skip the radioactive part in the title anymore. There only a few remaining technical details to be corrected prior to publication.

1) Based on the authors' replies, it seems that they are calculating and using the volume linear depolarization ratio in their analysis that corresponds to the total depolarization during scattering from a volume of air that includes both aerosols and particles. Please stick to the official nomenclature and use the term volume linear depolarization (VLDR) in the manuscript instead of just linear depolarization ratio.

Response:
The authors have specifically stated that the depolarization ratio presented in the manuscript is the linear volume depolarization ratio (L. 105-106). The authors have also included 3 references (Freudenthaler et al., 2009; Comerón et al., 2018).

2) The authors must also be careful when comparing their measured VLDR with values from the literature. The values found in the lidar literature correspond to the particle (aerosol) linear depolarization ratio (PLDR), that is the depolarization ratio after removing the influence of the molecules.
The authors must mention this in the manuscript, when they are directly comparing values with the literature. They must also keep in mind that the PLDR is generally larger than the VLDR because the molecular lin. dep. ratio is small (~0.004 at 355 with an IF ~ 0.5nm). Aerosol dominant scenes correspond to VLDR values closer to PLDR.

Response:
The authors have checked the references that were presented in the manuscript and have emphasized the type of depolarization ratio presented in their results (L. 150-154).

3) Please also provide a reference and/or a short description that explains the polarization calibration process.
Response:

The authors have already provided the references on the polarization calibration process in the manuscript (Burton et al., 2015 and Xu et al., 2022). The authors added the statement "The gain is obtained by measuring the signals from each channel when the polarizer is oriented at the parallel and perpendicular positions." (L. 108-109)

4) Please have a look at the pdf attachment that contains some suggestions for better phrasings
Response:
The authors couldn't find the pdf attachments. However, the authors believe that the reviewer is referring to the comments after comment #6.

5) In section 3.3, Skynet is mentioned. Is this a PredePom Instrument. Please mention whether the instrument is a sunphotometer or an in-situ instrument like a particle counter.
Response:
The authors have included the name of the instrument used by the Skynet program which is the Prede sky radiometer. The authors have added the statement "The SKYNET network uses the PREDE POM-01 and POM-02 as in-situ sky radiometers." (L 120-121)

6) Table1: Beam divergence: Please specify whether this is the beam divergence prior to expansion and also specify more details.
Is it the half or full angle beam divergence? Which part of the laser beam does it encircle e.g. 2 sigma (87%) or at 3 sigma (99%)
Response:
The beam divergence is the full angle divergence before expansion. This information can be obtained from the Spectra Physics website (https://www.spectra-physics.com/en/f/explorer-one-compact-laser). The authors have added the phrase "with a full beam divergence of 3 mrad prior to expansion". (L. 93-94)

Typos and rephrasing:
Response: The authors thanks the reviewer for noticing these typographical errors and for recommending better phrasing. The authors have corrected these and have responded to the reviewer's comments:

L29: " that can be left alone and gather data " --> that can perform continuous and automated measurements, gathering data
Response:
Done (L. 29-30)
L35: " This possibility is the focus explored in this paper." --> Such applications are the main focus of this paper.
Response:
Done (L. 35)
L86: " in analyzing" --> when analyzing
Response:
Done (L. 86)
L104: " The linear depolarization ratio" --> "The volume linear depolarization ratio (VLDR)..."

Response:
Done
L110: " is around 74 m horizontally and 19 m" --> is install at around 79 m horizontally and at 19 m vertically?...
Response:
Done (L. 112)
L192: remove the duplicate
Response:
Done (L. 194)

L219: Remove the duplicate
Response:
Done (L. 221)
L299: Add a space
Response:
Done (L. 293-297 and L. 302)

Reviewer 2:
Please fix Line 193 "This result is expected this result is expected"
Response:
Done (L. 194)

Paragraph starting on Line 239, Please fix figure 4 references, you states 4e and f but I think you mean g & h
Response:
Thank you for noticing this. The authors have made changes as shown in L. 242-243.

Line 290 do you mean "when the average extinction coefficient is compared with wind direction" instead of wind speed?
Response:
Thank you for noticing this. The authors have made changes as shown in L. 292.